# Norcantharidin Sensitizes Colorectal Cancer Cells to Radiotherapy via Reactive Oxygen Species–DRP1-Mediated Mitochondrial Damage

**DOI:** 10.3390/antiox13030347

**Published:** 2024-03-14

**Authors:** Qiong Xu, Heng Zhang, Haoren Qin, Huaqing Wang, Hui Wang

**Affiliations:** 1Department of Oncology, Tianjin Union Medical Center, Nankai University, Tianjin 300350, China; ezxxuqiong02@163.com (Q.X.); zhangheng@umc.net.cn (H.Z.); wanghuaqing@umc.net.cn (H.W.); 2School of Medicine, Nankai University, Tianjin 300350, China; qinhaoren@umc.net.cn; 3School of Integrative Medicine, Tianjin University of Traditional Chinese Medicine, Tianjin 301600, China

**Keywords:** ROS, NCTD, colorectal cancer, mitochondria, DRP1

## Abstract

Norcantharidin (NCTD), a cantharidin derivative, induces ROS generation and is widely used to treat CRC. In this study, we clarified the role and mechanism of action of norcantharidin in increasing CRC sensitivity to radiotherapy. We treated the CRC cell lines LoVo and DLD-1 with NCTD (10 or 50 μmol/L), ionizing radiation (IR, 6 Gy), and a combination of the two and found that NCTD significantly inhibited the proliferation of CRC cells and enhanced their sensitivity to radiotherapy. NCTD induced ROS generation by decreasing the mitochondrial membrane potential, increasing mitochondrial membrane permeability, and promoting cytochrome C release from mitochondria into the cytoplasm. IR combined with NCTD induced ROS production, which activated the mitochondrial fission protein DRP1, leading to increased mitochondrial fission and CRC sensitivity to radiotherapy. NCTD also reduced CRC cell resistance to radiotherapy by blocking the cell cycle at the G2/M phase and decreasing p-CHK2, cyclin B1, and p-CDC2 expression. NCTD and IR also inhibited radiation resistance by causing DNA damage. Our findings provide evidence for the potential therapeutic use of NCTD and IR against CRC. Moreover, this study elucidates whether NCTD can overcome CRC radiation tolerance and provides insights into the underlying mechanisms.

## 1. Introduction

Colorectal cancer (CRC) is the third most aggressive cancer worldwide and represents a serious threat to human health [1]. In 2022, approximately 592,232 new CRC cases and 309,114 CRC-related deaths were reported in China [2]. Current treatments for CRC are based on surgical resection, chemoradiotherapy, and molecular targeted therapy, with neoadjuvant chemoradiotherapy combined with surgery being the standard care for patients with locally advanced CRC [3]. Despite improvements in the treatment of CRC through early screening, radical surgery, and comprehensive CRC treatment guideline optimization, the overall outcome for CRC patients remains unfavorable, with a 5-year survival rate of approximately 65% [1]. Radiotherapy (RT) plays an important role in the treatment of CRC. However, patients with CRC are less sensitive to adjuvant RT. Therefore, studying radiation tolerance mechanisms and enhancing RT are of great significance.

Cancer cells generate reactive oxygen species (ROS) for survival, proliferation, angiogenesis, and metastasis [4]. Studies have revealed that a lower level of intracellular ROS following radiation leads to malignant tumor tolerance to RT [5,6]. The regulation of intracellular ROS levels is critical for cellular homeostasis because cells respond differently to different levels of ROS [7]. Low ROS levels in cancer cells can modulate energy production efficiency and promote cell proliferation. IR substantially increases ROS levels [8]. ROS are critical mediators of the lethal effects of ionizing radiation (IR) [4]. High levels of ROS can induce cell senescence and death through oxidative damage to intracellular biomacromolecules (e.g., proteins, lipids, RNA, and DNA). NADPH oxidase, another important source of ROS, is activated by radiation exposure, thus leading to persistent oxidative stress [9]. Cancer cells must combat high ROS levels, especially during the early stages of tumor development, and their high antioxidant capacity regulates ROS to levels compatible with cellular functions [10]. Therefore, enhancing ROS production after exposure to IR and maintaining oxidative stress in tumor cells may be the keys to overcoming radiation tolerance and enhancing tumor radiosensitivity.

The structural and functional integrity of mitochondria are essential for maintaining cellular energy and metabolic homeostasis. The mitochondria-dependent apoptotic pathway is the main mode of radiation-induced cell death, and it is initiated by excessive ROS production [11]. High ROS levels can disrupt mitochondrial function and lead to dysfunctional mitochondrial oxidative phosphorylation, which results in the release of large amounts of ROS from mitochondria, further exacerbating the intracellular ROS load [12]. When cells undergo metabolic or environmental stress, mitochondrial fission and fusion maintain mitochondrial function by regulating mitochondrial morphology and structure. Because IR induces ROS generation, it leads to impaired mitochondrial fusion and fission kinetics, which further lead to a loss of mitochondrial integrity and function. Dynamin-related protein 1 (DRP1) plays a key role in mitochondrial division. DRP1-Ser616 activation-driven excessive mitochondrial fission exacerbates oxidative stress and apoptosis [13,14,15]. During early apoptosis, DRP1 is recruited from the cytoplasm to the outer mitochondrial membrane, which triggers mitochondrial division. The inhibition of DRP1 has been shown to inhibit mitochondrial division as well as caspase activation and cell death [16]. Taken together, mitochondria are both the main target of intracellular ROS after IR and the main source of radiation-induced ROS. Radiation increases ROS production, which promotes DRP1 protein phosphorylation, leading to impaired mitochondrial dynamics and aggravated mitochondrial damage, which promote each other to form a cycle that ultimately leads to cell death. Therefore, enhancing ROS-mediated mitochondrial damage may increase the sensitivity of malignant tumors to IR.

Cantharidin is a traditional Chinese medicine extracted from the dried bodies of the insect *Mylabris phalerata*. Norcantharidin (NCTD) is synthesized by removing the methyl group at the 1,2 position from cantharidin, and it has stronger anti-tumor properties and fewer side effects than cantharidin [17]. Several studies have shown that NCTD can induce apoptosis in prostate, gastric, breast, and other tumor cells, mainly by increasing ROS production via a mechanism similar to that of radiation-induced cell death. For example, Shen et al. showed that NCTD induces mitochondrial membrane potential (MMP) reduction and mitochondrial dysfunction and revealed that the accumulation of intercellular ROS can eventually induce apoptosis by causing DNA damage [18,19,20]. In addition, NCTD can directly affect the expression of Bax, Bcl-2, and caspase-3 and -9. Specifically, NCTD significantly increases the expression of caspase-3, -8, and -9, decreases the expression of caspase-4 and -12 and the anti-apoptotic proteins Bcl-XL and Mcl-1, and increases the expression of the pro-apoptotic protein Bak. However, studies on the increased radiation sensitivity of CRC due to NCTD are limited; therefore, exploring the role of NCTD in CRC radiation tolerance is of great clinical significance.

Therefore, this study clarified the role and mechanism of action of norcantharidin in increasing the sensitivity of CRC to radiotherapy. CRC cell lines were treated with NCTD, ionizing radiation (IR), or a combination of the two. In addition, to elucidate the underlying molecular pathway, we investigated DRP1, a protein central to mitochondrial fission, and treated cells with the mitochondrial division inhibitor Mdivi-1. Moreover, in vivo experiments were performed to further validate the safety and efficacy of this treatment strategy.

## 2. Materials and Methods

### 2.1. Cell Culture

The human colon cancer cell line DLD-1 was purchased from Procell Life Science & Technology Co., Ltd. (Wuhan, China) and the LoVo cell line was gifted by the Laboratory of Translational Medicine, Tianjin People’s Hospital, Tianjin, China. All cells were incubated in RPMI 1640 medium (Gibco, Thermo, Waltham, MA, USA) containing 10% fetal bovine serum (FBS) (Gibco, Thermo, Waltham, MA, USA), 1% penicillin, and 0.1% streptomycin (Thermo, Waltham, MA, USA) at 5% CO_2_ and 37 °C.

### 2.2. Cell Irradiation and Treatment

Sealed sterile cell culture plates or dishes were placed in the center of the irradiation chamber and exposed to a radioactive source using 6 MeV high-energy X-rays generated by a linear accelerator (Varian Clinic 21ES, Palo Alto, CA, USA) at a dose rate of 3 Gy/min. NCTD was purchased from Yuanye Biotechnology Co., Ltd. (Shanghai, China). NCTD powder was dissolved in dimethylsulfoxide (DMSO) to make a stock solution of 100 mmol/L and kept at –20 °C. Depending on the purpose of the experiment, the stock solution was diluted in complete 1640 medium to obtain the desired concentration. Working solutions were freshly prepared from standard solutions and used for cell culture.

### 2.3. Cell Viability Assays

The viabilities of the LoVo and DLD-1 cells were measured using a CCK8 assay (Solarbio, Beijing, China) according to the manufacturer’s instructions. Briefly, the cells were plated in 96-well plates at a density of 5000 (DLD-1) or 7000 (LoVo) cells/well. After the cells adhered to the walls, they were exposed to 100 µL of culture medium containing various concentrations of NCTD (10, 20, 40, 80, 120, or 160 µM). For the inhibitor experiment, pretreatment with the inhibitor (5 mM N-Acetyl Cysteine (NAC) and 5 µM Mdivi-1) (MCE, Monmouth juncyion, NJ, USA ) was performed for 2 h before the NCTD (10 or 50 µM) treatment. Then, 10 µL of CCK-8 reagent was added to each well after culturing for 24, 48, and 72 h. Absorbance values were measured at 450 nm using an enzyme-labeled instrument (Bio-Rad, Hercules, CA, USA).

### 2.4. Single-Cell Gel Electrophoresis Assay

The CRC cells were subjected to an alkaline comet assay (ELK Biotechnology, Wuhan, China). Briefly, cells suspended in PBS were collected at a density of 1 × 10^6^ cells/mL and then mixed with low-melting-point agarose gel, spread onto chamber slides, and allowed to cool. The slides were immersed in a lysis solution and incubated at 4 °C for 2 h. The slides were incubated for 20 min at room temperature (22–26 °C) in an alkaline electrophoresis buffer (200 mM NaOH and 1 mM EDTA). The slides were left at room temperature for 60 min to maintain the DNA under alkaline conditions, followed by electrophoresis at 25 V and 300 mA for 30 min. Subsequently, the slides were washed at least three times with 0.4 mM Tris–HCl (pH = 7.5) before they were stained with propidium iodide (PI) (10 μL) for 10 min. Comet tails were observed using a fluorescence microscope.

### 2.5. Flow Cytometric Analysis

A cell cycle assay kit (Yeasen, Shanghai, China) was used to determine the effects of IR and NCTD on the cycle of CRC cells. Briefly, CRC cells were cultured in 6-well plates, NCTD was added 2 h before irradiation, and the IR and combined groups were irradiated with 6 Gy X-rays. The cells were then cultured in an incubator for 24 h and digested with trypsin, and the cell precipitates were fixed with pre-cooled 70% ethanol at 4 °C overnight. The next day, 10 µL of PI and 10 µL of RNase A were added, and the cells were protected from light for 30 min.

An Annexin V/PI (Abbkine, Wuhan, China) apoptosis detection kit was used to determine the effects of IR and NCTD on the percentage of apoptotic cells. Briefly, cells were treated with NCTD (10 and 50 µM) and/or IR (6 Gy) for 48 h or pretreated with 5 mM NAC and 5 µM Mdivi-1 for 2 h, followed by the addition of NCTD and/or IR for 48 h. Finally, the cells were harvested and washed with PBS. Each sample was stained with 200 µL of 1× binding buffer containing 5 µL of Annexin V-FITC and 2 µL of PI, and then flow cytometry (BD Biosciences, San Jose, CA, USA) was performed. The above data was analyzed using Flow Jo software version 10.8.1. 

### 2.6. Western Blot Analysis

Cells were resuspended in a RIPA buffer (Solarbio, Beijing, China) containing 1% PMSF (Solarbio, Beijing, China) and a 1% protease inhibitor cocktail (Thermo, Waltham, MA, USA), lysed on ice for 10 min, broken using an ultrasonic crusher, and then centrifuged at 12,000× *g* for 15 min at 4 °C. The supernatant was collected, and a BCA assay kit (Beyotime, Shanghai, China) was used to determine the total protein concentration. The samples were then supplemented with 5× SDS-PAGE loading buffer (Solarbio, Beijing, China) and denatured in a Thermo shaker at 95 °C for 10 min. Western blotting was performed as described previously [21]. Briefly, proteins (30 µg/well) were separated by SDS-PAGE and transferred onto PVDF membranes, and then the membranes were blocked with 5% non-fat dry milk and incubated with the diluted antibody (Appendix A) overnight on a shaker at 4 °C. The next day, the primary antibody was recovered, and the PVDF membrane was washed three times with TBST. The membrane was then incubated with horseradish peroxidase-labeled secondary antibodies (CST, Danvers, MA, USA) for 1 h at room temperature and visualized using an enhanced chemiluminescence reagent (Millipore, Darmstadt, Germany), and chemiluminescence was detected using a gel imaging system (Tanon, Shanghai, China).

### 2.7. Detection of ROS

ROS were detected using a DCFH-DA kit (Beyotime, Shang hai, China) according to the manufacturer’s instructions. Briefly, cells were treated with IR and/or NCTD for 24 h and stained with DCFH-DA (10 µM) for 30 min. Green fluorescence was analyzed using flow cytometry. To detect mitochondrial ROS, the cells were incubated with a 5 µM MitoSOX^TM^ (Yeasen, Shanghai, China) working solution for 10 min at 37 °C and visualized under a fluorescence microscope.

### 2.8. MitoTracker^®^ Red CMXRos

To prepare a 1 mM stock solution, we added 94.1 µL of DMSO to 50 µg of lyophilized solid for dissolution and then diluted the solution to a working concentration of 200 nM using 1640 complete medium. The cells were treated with the MitoTracker^®^ Red CMXRos (CST, Danvers, MA, USA) working solution and incubated for 30 min at 37 °C in the dark. If no other staining was performed, the staining solution was aspirated, washed with PBS three times (for 5 min each), blocked, and observed under a confocal microscope. If other staining was performed, the cells were fixed with cold methanol at –20 °C for 15 min and then washed with PBS three times (for 5 min each).

### 2.9. Colony Formation Assay

LoVo and DLD-1 cells were inoculated in 35 mm dishes, and the number of inoculated cells was adjusted according to the different radiation doses as follows: 200 (0 Gy), 400 (2 Gy), 1500 (4 Gy), 6000 (6 Gy), and 12,000 (8 Gy) cells. The cells were then pretreated with NCTD for 2 h and exposed to the indicated doses of X-radiation (Gy/min). When colonies were visible in the dishes, they were fixed, stained, and counted using ImageJ software version win64. Clone formation and cell survival rates were calculated, and a single-hit multitarget model was used to fit the cell survival curve and calculate the sensitization enhancement ratio (SER).

### 2.10. Immunocytofluorescence Assay

Cells were cultured in 6-well plates with or without IR/NCTD treatment. Depending on the experimental design, cells were collected at 1.5 h or 48 h after irradiation, washed three times with PBS, and fixed with 4% paraformaldehyde for 30 min. Cells were then penetrated using 0.5% Triton X-100 for 30 min and blocked with 5% BSA at room temperature for 60 min. Next, the primary antibodies γ-H2AX (CST, 1:200, #80312) and DRP1 (CST, 1:50, #8570) were incubated overnight at 4 °C. Then, the cells were incubated with Alexa Fluor 488-labeled secondary antibodies (CST, 1:1000) in the dark for 60 min. After washing the cells, the nuclei were stained with 10 μg/mL DAPI for 10 min. Finally, images were obtained using a confocal microscope (Leica Stellaris 8, Mannheim, Germany).

### 2.11. Measurement of Mitochondrial Membrane Potential (JC-1 Fluorescent Probe Experiment)

The mitochondrial membrane potential assay kit (Yeasen, Shanghai, China) uses JC-1 as a fluorescent probe to rapidly and sensitively detect changes in the mitochondrial membrane potential in cells, tissues, or purified mitochondria. It can be used to detect mitochondrial health and early apoptosis. Briefly, CRC cells were seeded in 24-well plates at a suitable density and treated with NCTD (10, 50 μM), IR (6 Gy), NAC, or Mdivi-1. The culture medium was removed after 24 h, the JC-1 working solution was added, and the cells were incubated for 15 min at 37 °C. Subsequently, the cells were washed with pre-warmed PBS two times. Finally, a fluorescence microscope (Nikon, Tokyo, Japan) was used to observe changes in ΔΨm. In healthy cells, JC-1 monomers aggregate to form polymers, and mitochondria show strong red fluorescence (excitation at 550 nm; emission at 600 nm). In apoptotic or necrotic cells, JC-1 exists as a monomer, and mitochondria show strong green fluorescence (excitation wavelength: 485 nm; emission wavelength: 535 nm). The ratio of the red fluorescence signal to the green fluorescence signal was then calculated and used to determine cell health.

### 2.12. RNA Isolation and Quantitative Real-Time PCR

Total RNA was extracted using TRIzol (Invitrogen, Carlsbad, CA, USA) and used for cDNA synthesis, using a Hifair II 1st Strand cDNA Synthesis Kit (Yeasen, Shanghai, China). A qRT-PCR was performed using the qPCR SYBR Green Master Mix kit (Yeasen, Shanghai, China). Gene expression was normalized to GADPH. The PCR primers (Sangon Biotech, Shanghai, China) are shown in Appendix A.

### 2.13. Hematoxylin and Eosin Staining and Immunohistochemistry Assay

Tumor tissues were dehydrated, embedded, cut into slices, and stained with a hematoxylin and eosin (H&E) solution for 10–20 min. The slices were placed in 1% alcohol hydrochloric acid for differentiation for 1–3 s, and then the color changed back to blue. Neutral gum was used for sealing, and the slices were observed under a microscope (Nikon, Tokyo, Japan).

Paraffin sections were deparaffinized for 2 h at 65 °C, followed by rehydration in an alcohol series and sodium citrate buffer. The sections were covered with endogenous peroxidase and blocked with 5% normal goat or mouse serum. Sections were incubated with anti-Ki67 antibodies (1:400), anti-8OHDG antibodies (1:2400), anti-ϒ-H2AX antibodies (1:800), and anti-cleaved caspase-3 (1:1800) at 4 °C overnight. Immunohistochemical (IHC) staining was performed using HRP conjugates and diaminobenzidine. IHC images were captured using a microscope. Three independent random fields were selected for a semi-quantitative analysis using ImageJ software version win64, and the average optical density (AOD) was measured using the following formula: AOD = integrated optical density (IOD)/area.

### 2.14. Animals

Four-week-old male BALB/c nude mice were purchased from Beijing Vital River Laboratory Animal Technology (Beijing, China). All animal experiments were performed according to National Institutes of Health guidelines and Tianjin University of TCM protocol. The animals were housed in a temperature-controlled room (22 ± 2 °C) with a 12 h light/dark cycle and free access to standard mouse chow and water. The feeding conditions (food and water) were monitored daily. After 1 week of acclimatization, the mice were subcutaneously inoculated with 1 × 10^7^ DLD-1 cells in the left inguinal area. Approximately 10 days later, when the size of the tumor reached approximately 5 mm, the mice were randomly divided into four groups (5–6 mice) according to the experimental design. Tumor-burdened mice in the IR and combination groups were administered 10 Gy of X-ray irradiation at a time. Then, the combined group was continuously intraperitoneally injected with NCTD (5 mg/kg) for 14 days. The body weight of the mice was measured every 2 days, and tumor size was also measured every 2 days using digital calipers. At the end of the treatment, tumor tissues, livers, lungs, and kidneys were harvested for further study. Tumor volumes were calculated according to the following formula: volume = 0.5 × length × width^2^.

### 2.15. TUNEL Assay

Experiments were performed according to the manufacturer’s instructions (Yeasen, Shang hai, China). Tissue sections were routinely dewaxed and rehydrated in an alcohol series. Then, the tissue sections were covered with Proteinase K solution and incubated for 20 min at room temperature. The tissues were washed three times with PBS and incubated with the TUNEL reaction mixture for 60 min at 37 °C in a damp and dark place. Finally, the samples were observed under a fluorescence microscope.

### 2.16. Statistical Analyses

All experiments were performed in triplicate. The results are shown as mean ± standard deviation (SD) values. GraphPad Prism statistical software (V9) was used for data analysis. Data were analyzed using a one-way analysis of variance (ANOVA). The significance of the parameters was set at *p* < 0.05.

## 3. Results

### 3.1. NCTD Inhibits the Proliferation of CRC Cells and Enhances Their Radiosensitivity In Vitro

To test the anti-tumor effects of NCTD on CRC cells, we first examined cell viability after treatment with different concentrations of NCTD. The viability of the LoVo and DLD-1 cells was significantly inhibited at 24, 48, and 72 h after NCTD treatment in a dose- and time-dependent manner (Figure 1A–E). We used GraphPad Prism statistical software (V9) to calculate the IC20 values, which were 9.455 μM and 50.467 μM for LoVo and DLD-1, respectively. For the convenience of calculation and dosing, we used 10 and 50 μM as the subsequent drug concentrations. To investigate the effect of NCTD on CRC cell radiosensitivity, the cells were treated with NCTD at 10 and 50 μM, followed by IR at different doses. The inhibitory effect on colony formation was significantly enhanced in the combined group, and the survival fraction was significantly lower than that in the radiotherapy-only group. A SER > 1 suggests that a drug has a sensitizing effect. We calculated the SERs of the LoVo and DLD-1 cells using the single-hit multitarget model, and the values were 1.75 and 1.04, respectively (Figure 1F,G). To evaluate the inhibitory effect of NCTD in combination with IR on CRC cells, the two cell lines were subjected to different treatments, and colony formation was observed. The IR/NCTD groups effectively inhibited colony formation in CRC cells (Figure 1H–K).

### 3.2. NCTD Induces DNA Damage and Mitochondria-Dependent Apoptosis in CRC Cells

Damage to DNA double strands by energetic particle beams is the main cause of cell death by IR and a direct cause of radiosensitization. We explored whether IR combined with NCTD affects DNA damage induced by IR. To this end, we assessed phosphorylated H2AX (γ-H2AX) levels in CRC cells using immunofluorescence. In cells treated with NCTD alone, the level of γ-H2AX was remarkably higher than in the control cells, while in cells treated with IR/NCTD, the level of γ-H2AX was significantly higher than that in cells treated with IR alone (Figure 2A,B,D). These results were further confirmed by Western blotting (Figure 2C). In addition to the immunofluorescence assay, single-cell gel assays were used to detect DNA damage. An analysis of the percentage of tail moments revealed negligible comet tail formation in cells treated with NCTD alone compared to cells in the control group. As predicted, the combined treatment led to a higher percent of tail DNA and longer tail length than IR alone, further exacerbating DNA damage (Figure 2E−G).

To determine whether the radiosensitizing effect of NCTD was related to apoptosis, the apoptosis rate of the cells was measured using flow cytometry after double staining with Annexin V-FITC and PI. In addition, the expression of Bcl-xl, Bax, Bim, survivin, and cleaved caspase-3 was detected by Western blotting. After 48 h, the apoptosis rate was significantly increased in the IR/NCTD group compared to the IR alone group (Figure 2H). As shown in Figure 2H, the total apoptosis rate increased from 23.77% to 31.91% in LoVo cells after treatment with 10 µM NCTD for 48 h compared with that of the IR groups. The same results were obtained for the DLD-1 cell line. Moreover, in the IR/NCTD group, the expression of Bcl-xl and survivin was decreased while that of Bax, Bim, and cleaved caspase-3 was increased (Figure 2J–K). Cytochrome C (Cyto C) release is a key step in the mitochondrial apoptotic pathway, and the expression of Cyto C was increased in the IR/NCTD group compared to the IR alone group (Figure 2I,L).

### 3.3. NCTD Induces CRC Cell Senescence and Blocks the Cell Cycle

In addition to apoptosis, cellular senescence is an effective mechanism of inhibiting tumor cell proliferation and survival. We used a senescence β-galactosidase staining kit and found that compared to the IR group, the IR/NCTD combination group produced significantly more dark blue products (catalyzed by senescence-specific β-galactosidase) and that the activity level of senescence-associated β-galactosidase (SA-β-gal) was elevated (Figure 3A–C). In addition, cells treated with IR/NCTD showed significantly higher levels of expression of P21 and P16 than IR alone (Figure 3D–F).

The proportion of cells in the G2/M phase significantly increased in the IR/NCTD combination group (Figure 3G,H). The cell-cycle-related proteins cyclin B1, p-CDC2, and p-CHK2 were significantly decreased in the IR/NCTD combination group compared with the IR group (Figure 3I,J).

### 3.4. NCTD Combined with IR Impairs Mitochondrial Morphology and Function, Leading to Increased Mitochondrial Division

We further investigated the potential mechanisms by which NCTD sensitizes CRC cells to IR. Because apoptosis is induced through the mitochondrial pathway, we examined the effects of IR combined with NCTD on mitochondrial morphology and function. We used MitoTracker^®^ to specifically label the IR group, NCTD group, and IR/NTCD combination group. The results revealed that in the control group, the mitochondria were mostly “rod-like”, while in the combined group, mitochondrial fragmentation was markedly increased and punctate compared to that in the IR group (Figure 4A).

Since mitochondria show significant morphological changes in response to treatment with IR and NCTD, we sought to determine whether mitochondrial functions were also impaired. The loss of MMP is an important indicator of mitochondrial dysfunction. Therefore, we used JC-1 to measure MMP. In apoptotic cells, JC-1 cannot cross the mitochondrial membrane and therefore cannot form JC-1 aggregates, thus maintaining its original green fluorescence. Consistent with the expected results, green fluorescence was significantly increased in the IR/NCTD group compared to the IR group (Figure 4B).

In a further study, we found that IR combined with NCTD resulted in the activation of DRP1, a key regulator of mitochondrial division [22]. DRP1 is the most important protein involved in mitochondrial fission because it provides the energy necessary for fission [23]. The phosphorylation of DRP1 at serine 616 and activation induce additional mitochondrial division [24]. Here, we found that the level of phosphorylated DRP1 was significantly increased in the IR/NCTD group (Figure 4C). Similarly, the mRNA level of DRP1 was increased (Figure 4E,F).

To further investigate changes in mitochondrial dynamics, we evaluated the expression levels of mitochondrial fission- and fusion-related proteins. In the IR/NTCD group, the expression of MFF, Mifn1, Mifn2, and OPA1 was reduced in both LoVo and DLD-1 cells (Figure 4C,D), suggesting that NCTD combined with IR disrupts mitochondrial fission and fusion. DRP1 is a cytoplasmic protein that can induce division upon its transfer to mitochondria. IR causes DRP1 to transfer from the cytoplasm to the mitochondria and co-localize with the mitochondria, while NCTD combined with IR increases the translocation of DRP1 into mitochondria (Figure 4G,H).

### 3.5. NCTD Increased CRC Cell Apoptosis via Upregulating ROS Levels

Next, we investigated the potential mechanisms by which NCTD sensitizes CRC cells to IR. Radiobiological studies have shown that conventional doses of X-rays do not directly kill tumor cells but cause persistent damage to DNA by ROS stimulation, resulting in single- or double-stranded breaks in cancer cells. They may also cause ROS-induced lipid peroxidation, as well as ROS damage to mitochondria and the endoplasmic reticulum, leading to tumor cell death [10]. To determine whether NCTD induces an increase in ROS levels after IR, we measured intracellular and mitochondrial ROS levels using the fluorescent indicator DCFH-DA. The IR/NCTD combination group showed significantly increased intracellular (Figure 5A,C) and mitochondrial (Figure 5B,D) ROS levels. We further used NAC, a ROS scavenger, to identify whether ROS mediate the anticancer properties of NCTD and found that NAC rescued the inhibition of clone formation caused by combination therapy (Figure 5E) and inhibited intracellular (Figure 5F,G) and mitochondrial (Figure 5H–J) ROS generation. We then investigated the effect of NAC on apoptosis and showed that NAC reversed IR- and NCTD-induced apoptosis, resulting in a significant decrease in the apoptosis rate (Figure 5K–N), which was verified by Western blotting (Figure 5O,P).

### 3.6. ROS Lead to Excessive Mitochondrial Division by Increasing Mitochondrial Damage in CRC Cells

ROS production and mitochondrial damage are mutually inducing and represent common mechanisms for many apoptosis-inducing factors. Therefore, to explore the role of ROS in mitochondrial damage, mitochondrial changes were examined in NAC-treated CRC cells. We found that NAC pretreatment significantly reversed the combination-treatment-mediated decrease in ΔΨm, with a significant decrease in green fluorescence and an increase in red fluorescence after NAC treatment (Figure 6A,B). We also observed changes in mitochondrial morphology, which was more punctate and fragmented in the combined treatment group and reversed by NAC, with a more rod-like distribution (Figure 6C).

DRP1 activation can be attributed to mitochondrial ROS accumulation [25,26]. To further explore the role of ROS in mitochondrial fission and fusion, we pretreated cells with NAC (an ROS inhibitor) and found that the NAC scavenging of mitochondrial ROS led to a decrease in DRP1 activation, the inhibition of mitochondrial fission, and an increase in the expression of the mitochondrial fusion proteins Mifn1, Mifn2, and OPA1 (Figure 6D). In addition, confocal microscopy showed that NCTD combined with IR induced the co-localization of DRP1 with mitochondria, which was reversed in the NAC-treated group (Figure 6E). The scavenging of ROS by NAC resulted in reduced DRP1 activation and a concomitant inhibition of mitochondrial fission.

### 3.7. DRP1 Mediates IR/NCTD-Induced Mitochondrial Division and Inhibition of DRP1 Expression Rescues Mitochondria-Dependent Apoptosis and Decreases ROS

DRP1 plays a key role in mitochondrial division and may mediate mitochondrial fusion and division [23,27]. To further confirm the role of DRP1 in mitochondrial division, we pretreated cells with Mdivi-1 and observed the role of DRP1 in mitochondrial damage. First, we used confocal microscopy to observe the mitochondrial morphology and found that the mitochondria were swollen and fragmentation was increased after co-treatment; however, these changes were reversed by Mdivi-1. In addition, Mdivi-1 inhibited mitochondrial division and reduced mitochondrial damage, ultimately alleviating mitochondrial dysfunction (Appendix A). Mdivi-1 did not significantly alter the expression of total DRP1 but effectively alleviated the expression of phosphorylated DRP1 and reversed the reductions in OPA1, MFF, Mifn1, and Mifn2 in CRC cells induced by the combined IR and NCTD treatment (Figure 7C–E). The latter induced the translocation of cytoplasmic DRP1 to the mitochondria; however, no translocation occurred in CRC cells treated with Mdivi-1 (Figure 7A,B).

To elucidate the relationship between mitochondrial fission and apoptosis, we pretreated the cells with Mdivi-1. Numerous studies have shown that the inhibition of DRP1 delays Cyto C release and inhibits apoptosis in vivo [28,29,30]. We investigated whether mitochondrial division promotes apoptosis and found that cells pretreated with Mdivi-1 showed a significant decrease in apoptosis (Figure 7G), a decrease in the expression of the pro-apoptotic proteins Bax and cleaved caspase-3, an increase in the expression of Bcl-xl (Figure 7H–J), and a significant increase in cell viability (Figure 7F). We then examined changes in ROS levels after Mdivi-1 treatment and found that ROS production was inhibited (Figure 7K). The results of previous studies have shown that NAC reverses the expression of DRP1, suggesting that ROS and DRP1 promote each other’s expression.

### 3.8. NCTD Increased the Radiosensitivity of CRC Cells In Vivo

Tumor size and number were significantly reduced in mice treated with NCTD and IR compared with those treated with either modality alone (Figure 8A–E). None of the mice showed mortality or significant weight loss during the treatment period (Figure 8F,G), and H&E staining results showed no apparent toxicity to the heart, liver, or kidney in mice treated with NCTD and IR, suggesting the safety of this combination (Figure 8J). The TUNEL assay results showed that NCTD combined with IR resulted in increased apoptosis compared with the other treatment groups (Figure 8H). IHC staining showed a significant decrease in the Ki67-positive cell population, an increase in cleaved caspase-3 expression, and a significant increase in the expression of 8-OHDG and γ-H2AX in tumors co-treated with IR and NCTD (Figure 8I,K). This suggests that the tumor tissues were in a state of oxidative stress after the co-treatment and thus showed a reduced proliferative capacity and increased apoptosis rate.

## 4. Discussion

Adjuvant radiotherapy has obvious advantages in reducing tumor downstaging, increasing the anal preservation rate, and prolonging tumor-free survival, thereby greatly improving the quality of life of patients with rectal cancer. Therefore, the NCCN treatment guidelines in the United States have proposed adjuvant RT combined with surgical treatment as the recommended treatment for locally advanced rectal cancer [3,31]. However, sensitivity to adjuvant RT is generally low in patients with rectal cancer. The results of the German CAO/ARO/AIO-94 trial showed that the rate of complete regression after preoperative RT for rectal cancer was only 8% [31], indicating that a large proportion of patients do not benefit from adjuvant RT. Consequently, numerous studies have been conducted to develop effective treatment strategies for patients with CRC to improve survival rates and reduce mortality.

Recently, the role of intracellular ROS levels in resistance to RT has received increasing attention. Radiobiology has shown that conventional doses of high-energy X-rays do not directly kill tumor cells but rather stimulate ROS-sustained damage to DNA, thereby destroying single- or double-stranded DNA. Moreover, ROS causes protein and lipid peroxidation and the destruction of mitochondria and the endoplasmic reticulum, which together lead to tumor cell death [10].

NCTD is widely used for the clinical treatment of CRC and other malignant tumors and acts as a chemosensitizer in various cancer cell lines [19,20]. NCTD has been shown to inhibit the expression of Ki-67 and Bcl-2, induce S-phase cell-cycle arrest, and directly or indirectly downregulate the VEGF-A,-C,-D/VEGFR-2,-3 signaling pathways, thereby inhibiting the proliferation of CRC cells [32]. Qiu et al. showed that NCTD inhibits cell growth by inhibiting the expression and phosphorylation of EGFR and c-Met in human colon cancer cells. Another study showed that NCTD inhibits tumor angiogenesis by blocking the VEGFR2/MEK/ERK signaling pathway [33]. In addition, NCTD inhibits the epithelial–mesenchymal transition in colon cancer cells by inhibiting the alphavbeta6-ERK-Ets1 signaling pathway [34]. However, few studies have focused on the increased sensitivity of NCTD to radiotherapy in rectal cancer, and the mechanism underlying the ability of NCTD to overcome radiation tolerance in rectal cancer has not been reported. Therefore, we sought to determine whether NCTD could increase CRC radiotherapy sensitization by increasing ROS production.

Ataxia-telangiectasia mutated (ATM) protein, a member of the phosphatidylinositol 3-kinase-related kinases family, induces a break in the cellular DNA double-strand. In addition, ATR (ATM and RAD 3-related) and other proteins phosphorylate and modify serine 139 on H2AX to form phosphorylated H2AX (γ-H2AX) [35]. In response to double-stranded breaks, the histone H2AX is phosphorylated at serine 139 in a region of several mega-base pairs, thus forming discrete nuclear foci that can be detected using a fluorescence microscope. Our results showed that the level of γ-H2AX was significantly elevated in the combined group in both cell lines, as evidenced by enhanced fluorescence in the cell nuclei. The involvement of cell-cycle blocking and senescence in IR tolerance and tumor metastasis has been previously reported [36]. Our results indicate that NCTD inhibits tumor cell survival by inducing cell-cycle arrest and senescence. This phenomenon indicates that NCTD could reverse the RT-tolerant state of CRC cells.

Our study showed that NCTD increases the sensitivity of CRC cells to RT by inducing apoptosis via ROS-DRP1-mediated mitochondrial damage. Mitochondria are the main sources and targets of intracellular ROS after exposure to radiation. IR induces ROS production and disrupts mitochondrial fusion and fission, leading to a loss of mitochondrial integrity and function. An NCTD-induced MMP reduction was reversed by ROS removal, suggesting that ROS contribute to NCTD-induced mitochondrial damage. Further experiments revealed that ROS-induced mitochondria-dependent apoptosis showed an increase in the expression of Cyto C, a decrease in the expression of the anti-apoptotic proteins Bcl-xl and survivin, and an increase in the expression of the pro-apoptotic proteins Bax, Bim, and cleaved caspase-3, and these changes were partially reversed by NAC, a typical antioxidant [37]. These results confirm that ROS are important mediators of radiation-induced damage.

ROS impair mitochondrial dynamics, leading to imbalances in mitochondrial fusion and division. Therefore, we further explored mitochondrial fission and fusion in ROS-promoted mitochondrial injury and found that IR combined with NCTD increased the expression of the mitochondrial fission protein DRP1 and decreased that of mitochondrial fusion proteins (mifn1, mifn2, and OPA1). NAC partially restored the expression of the mitochondrial fusion and fission proteins, implying that ROS play an important role in mitochondrial fusion and division. To elucidate the underlying molecular pathway, we investigated DRP1, a protein central to mitochondrial fission, and found that IR combined with NCTD significantly upregulated DRP1 phosphorylation, leading to its activation, and subsequently increased mitochondrial division [23]. Studies have shown that mitochondrial division promotes apoptosis [38]. Moreover, the inhibition of mitochondrial fusion has been shown to promote apoptosis [27,28,29]. To further elucidate the relationship between apoptosis and mitochondrial division, we treated cells with the mitochondrial division inhibitor Mdivi-1, which led to a significant decrease in apoptosis and mitochondrial damage, suggesting that increased mitochondrial division induces apoptosis.

To further elucidate the relationship between ROS generation and mitochondrial division, we pretreated the cells with Mdivi-1 and measured intracellular ROS levels. Mdivi-1 partially inhibited the expression of ROS. Early in apoptosis, DRP1 is recruited from the cytoplasm to the outer mitochondrial membrane to trigger mitochondrial fission [39]. DRP1 induces apoptosis by stimulating Bax oligomerization and massive Cyto C efflux to activate the apoptotic protease activator (Apaf-1), thereby irreversibly altering mitochondrial function [39]. Considering that NAC inhibited the expression of DRP1, these results suggest that ROS and DRP1 jointly promote apoptosis. However, in this study, IR directly led to ROS generation, which initiated mitochondrial damage-induced apoptosis. Previous studies have shown that NCTD facilitates the accumulation of ROS and activates the mitochondrial pathway to induce tumor cell apoptosis [40,41]. Consistent with previous findings, our results suggest that IR in combination with NCTD induces mitochondrial damage by increasing ROS production, thereby promoting apoptosis. Thus, these results reveal a potential mechanism for the radiosensitivity enhancement effect of NCTD, Figure 9 shows the mechanism of action of NCTD to improve the sensitivity of colorectal cancer radiotherapy.

## 5. Conclusions

NCTD combined with IR increased the sensitivity of CRC cells to RT by mediating the ROS-driven activation of DRP1 and inducing mitochondrial division and apoptosis. In vivo experiments further validated the safety and efficacy of this treatment strategy.

## Figures and Tables

**Figure 1 antioxidants-13-00347-f001:**
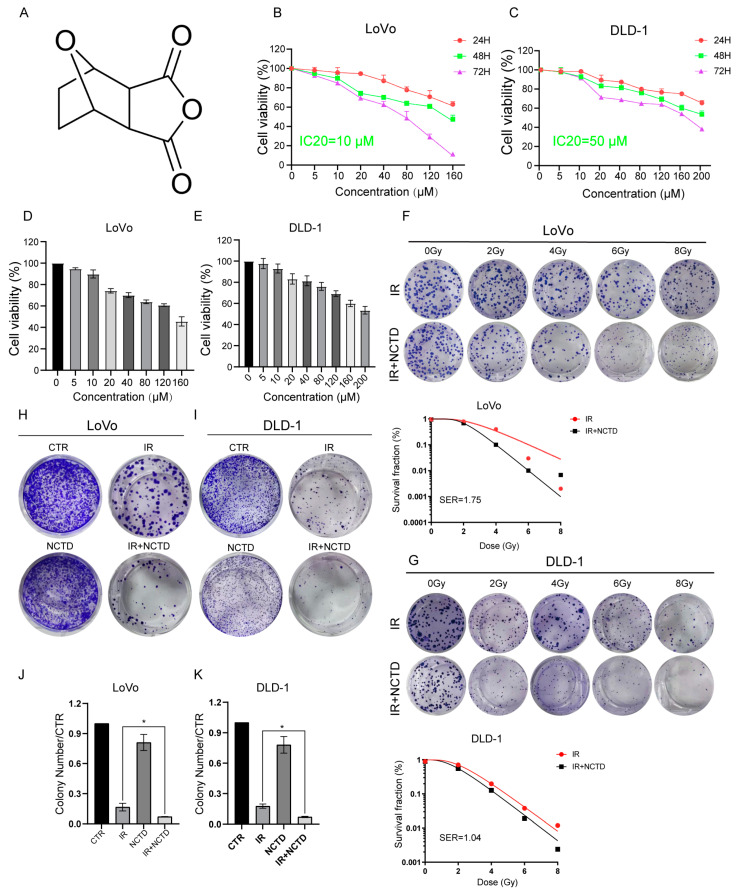
Norcantharidin (NCTD) enhances colorectal cancer (CRC) cell radiosensitivity in vitro. (**A**) The chemical structure of NCTD. (**B**,**C**) The viability of LoVo and DLD-1 cells treated with NCTD. (**D**,**E**) CRC cell viability and IC20 values after 48 h of treatment with different concentrations of NCTD. (**F**,**G**) Colony formation assay showing the enhanced radiosensitivity of CRC cells in vitro, as evaluated by the radiation multi-target single-hit model, after NCTD (10 and 50 µM) treatment. (**H**–**K**) CRC cells were treated with ionizing radiation (IR, 6 Gy) alone or in combination with NCTD (10 and 50 µM) for cell clone formation, *, *p* < 0.5.

**Figure 2 antioxidants-13-00347-f002:**
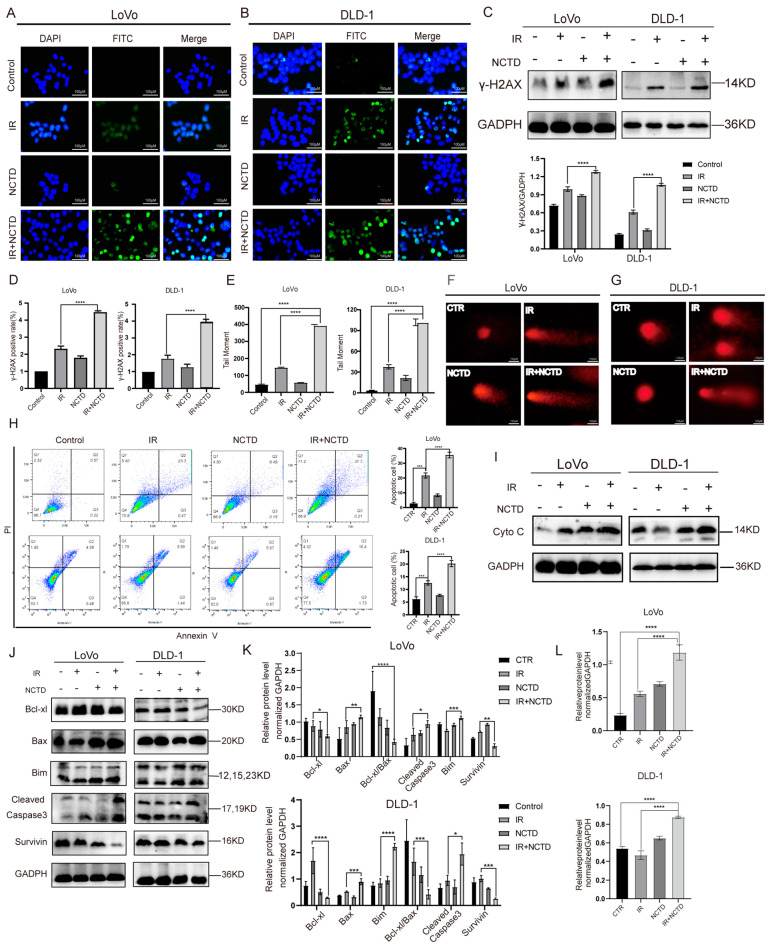
NCTD induces DNA damage and mitochondria-dependent apoptosis in CRC cells. (**A**,**B**) Fluorescence microscopy images showing increased DNA fracture (ϒ-H2AX, green) after the combination treatment of IR (6 Gy) and NCTD (10 and 50 µM). (**C**) Representative immunoblotting of DNA damage (ϒ-H2AX) in CRC cells. ****, *p* < 0.0001. (**D**) Histogram of intracellular ϒ-H2AX-positive cytometry after IR combined with NCTD treatment. ****, *p* < 0.0001. (**E**) Casplab software version 1.2.3 b1 was applied to process and analyze the comet images, and the lengths of the tail moments of the four groups of comets after different interventions were analyzed. ****, *p* < 0.0001. (**F**,**G**) Detection of DNA damage using the comet assay. Scale bar = 100 µm. (**H**) NCTD increased apoptosis at 48 h post IR in CRC cells. ***, *p* < 0.001, ****, *p* < 0.0001 (**I**) NCTD increased the expression of Cyt C 48 h post IR. (**L**) Histogram of Cyt C protein expression after different treatments. ****, *p* < 0.0001. (**J**,**K**) Expression of Bcl-xl, Bax, Bim, cleaved caspase-3, and survivin was detected by Western blotting 48 h post IR in CRC cells. *, *p* < 0.05; **, *p* < 0.01, ***, *p* < 0.001, ****, *p* < 0.0001.

**Figure 3 antioxidants-13-00347-f003:**
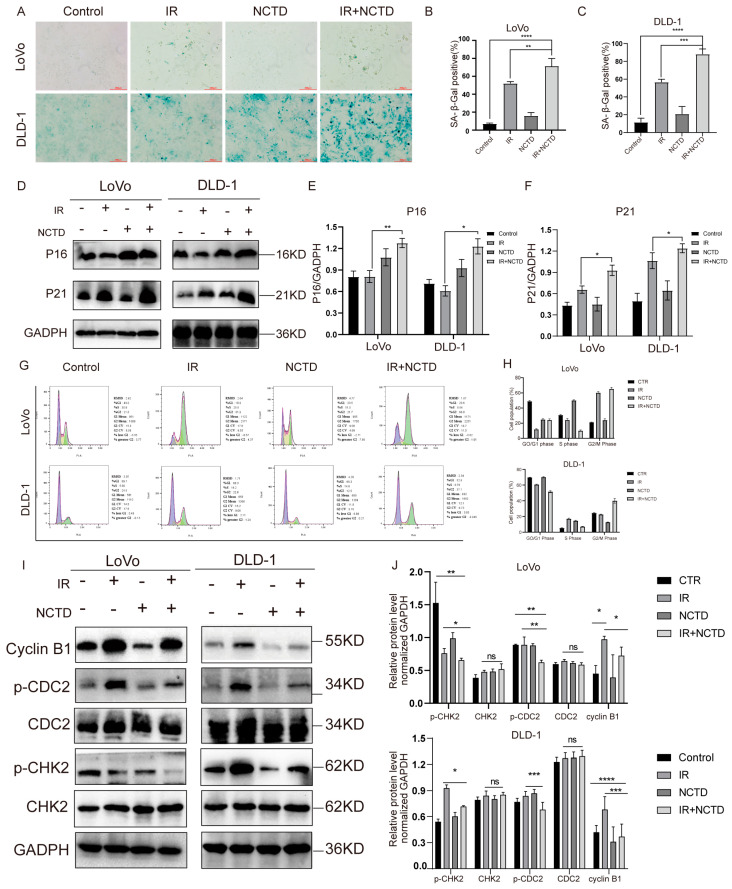
NCTD induces CRC cell senescence and blocks the cell cycle post IR. LoVo and DLD-1 cells were treated with NCTD for 24 or 48 h. (**A**–**C**) NCTD increased the expression of senescence-associated β-galactosidase (SA-β-gal) after IR in CRC cells. **, *p* < 0.01, ***, *p* < 0.001, ****, *p* < 0.0001. Scale bar = 100 µm. (**D**–**F**) Western blotting analyses of p21 and p16 in irradiated LoVo and DLD-1 cells. *, *p* < 0.05; **, *p* < 0.01. (**G**,**H**) The CRC cell cycle was assessed using flow cytometry. (**I**) Representative immunoblotting of the cell-cycle proteins cyclin B1, CDC2, p-CDC2, CHK2, and p-CHK2. (**J**). Histogram of p21 and p16 proteins in gray values. *, *p* < 0.05; **, *p* < 0.01, ***, *p* < 0.001, ****, *p* < 0.0001, ns, *p* > 0.05.

**Figure 4 antioxidants-13-00347-f004:**
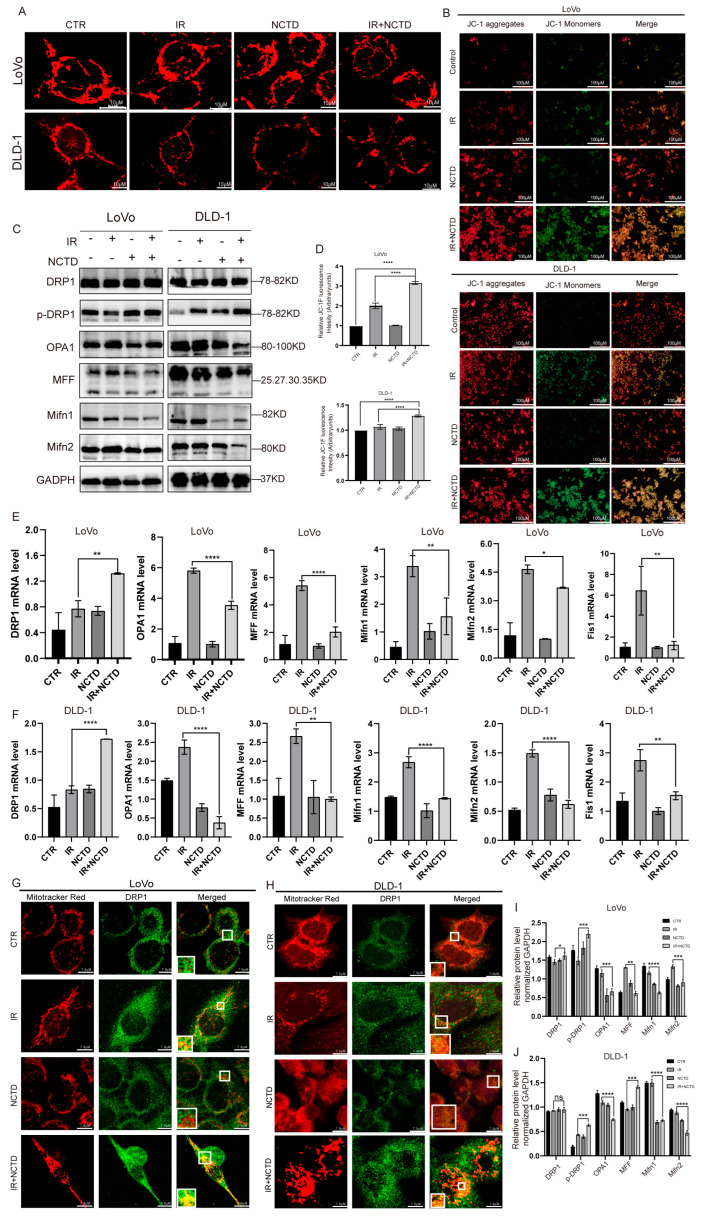
Effect of IR combined with NCTD on mitochondrial damage and mitochondrial division in CRC cells. (**A**) Morphology of mitochondria in CRC cells treated with IR and/or NCTD, as determined by confocal microscopy. Scale bar = 10 µm. (**B**) JC-1 in CRC cells. Scale bar = 100 µm. (**C**) Representative Western blot results and the quantification of mitochondrial fusion- and fission-associated proteins. (**D**) JC-1 fluorescent probe bar chart, ****, *p* < 0.0001. (**E**,**F**) Statistical analysis of the mRNA levels of mitochondrial fission- and fusion-associated proteins in CRC cells. *, *p* < 0.05, **, *p* < 0.01, ****, *p* < 0.0001. (**G**,**H**) Representative confocal images showing DRP1 immunofluorescence and mitochondria stained with MitoTracker^®^. Nuclei were stained with DAPI. Scale bar = 10 µm. (**I**,**J**) The quantification of mitochondrial fusion- and fission-associated proteins. *, *p* < 0.05, **, *p* < 0.01; ***, *p* < 0.001, ****, *p* < 0.0001, ns, *p* > 0.05.

**Figure 5 antioxidants-13-00347-f005:**
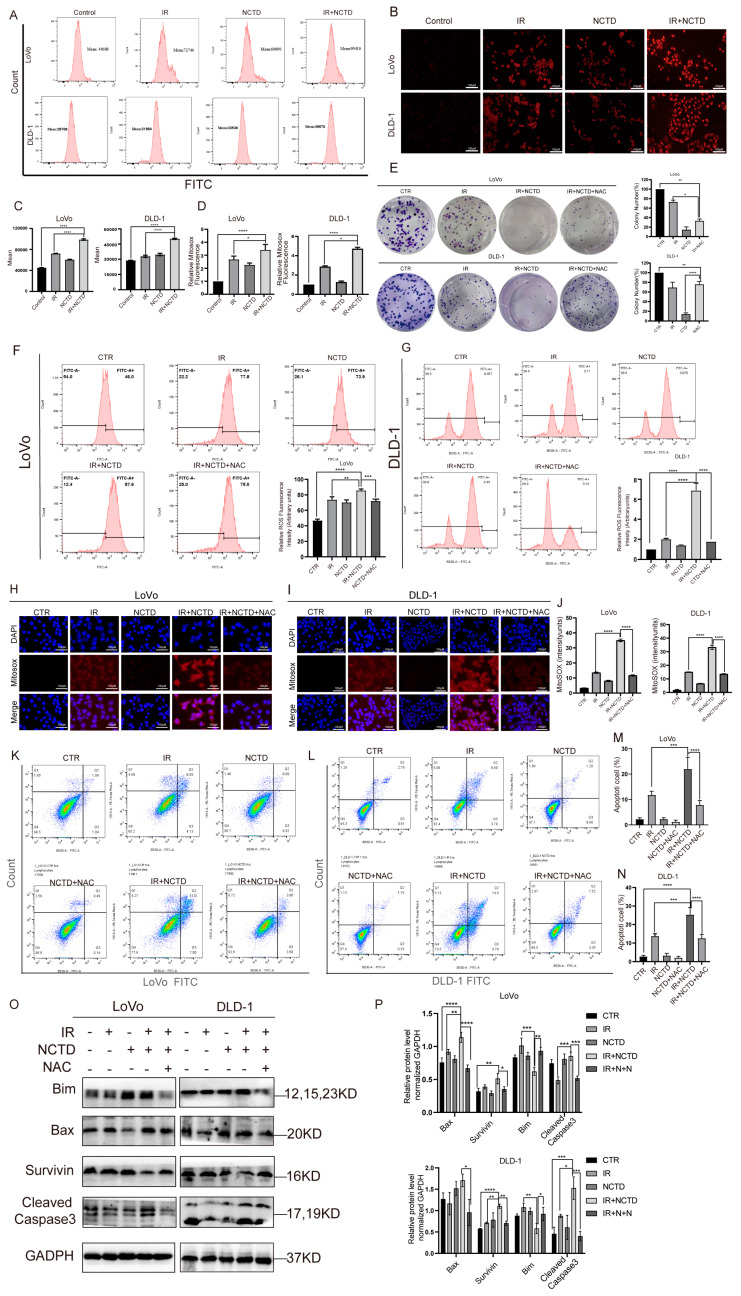
NCTD increased CRC cell apoptosis by upregulating ROS levels. (**A**−**D**) Combination therapy increased intracellular and mitochondrial ROS levels. *, *p* < 0.05, ****, *p* < 0.0001. (**E**) Images of clone formation in the groups after the addition of NAC. *, *p* < 0.05; **, *p* < 0.01, ****, *p* < 0.0001. (**F**,**G**) NAC decreased intracellular ROS levels in CRC cells. **, *p* < 0.01, ***, *p* < 0.01, ****, *p* < 0.0001. (**H**−**J**) NAC also inhibited ROS generation in mitochondria. ****, *p* < 0.0001. (**K**,**L**) Detection of apoptosis by flow cytometry after Annexin V–propidium iodide (PI) staining. (**M**,**N**) Annexin V−PI staining histograms. ***, *p* < 0.01, ****, *p* < 0.0001. (**O**,**P**) Representative immunoblotting of related apoptosis proteins and statistical histograms. *, *p* < 0.05; **, *p* < 0.01, ***, *p* < 0.01, ****, *p* < 0.0001.

**Figure 6 antioxidants-13-00347-f006:**
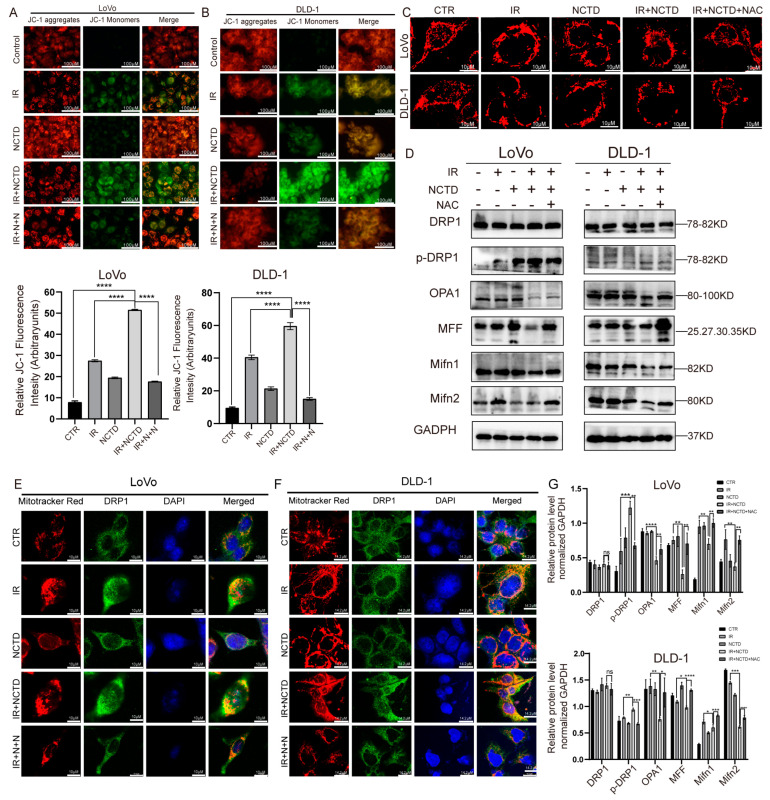
Reactive oxygen species (ROS) lead to excessive mitochondrial division by increasing mitochondrial damage in CRC cells. (**A**,**B**) Mitochondrial membrane potential in CRC cells. **** *p* < 0.0001. (**C**) Mitochondrial morphology, as observed using confocal microscopy. NAC reverses mitochondrial fragmentation caused by combination therapy. Scale bar = 10 µm. (**D**) Representative immunoblotting of mitochondrial fusion− and fission−associated proteins after NAC treatment in CRC cells. (**E**,**F**) Immunofluorescence of DRP1 and mitochondria after NAC treatment in CRC cells. Scale bar = 10 µm. (**G**) Statistical bar graph of mitochondrial division and fusion. *, *p* < 0.05; **, *p* < 0.01, ***, *p* < 0.001, ****, *p* < 0.0001, ns > 0.05.

**Figure 7 antioxidants-13-00347-f007:**
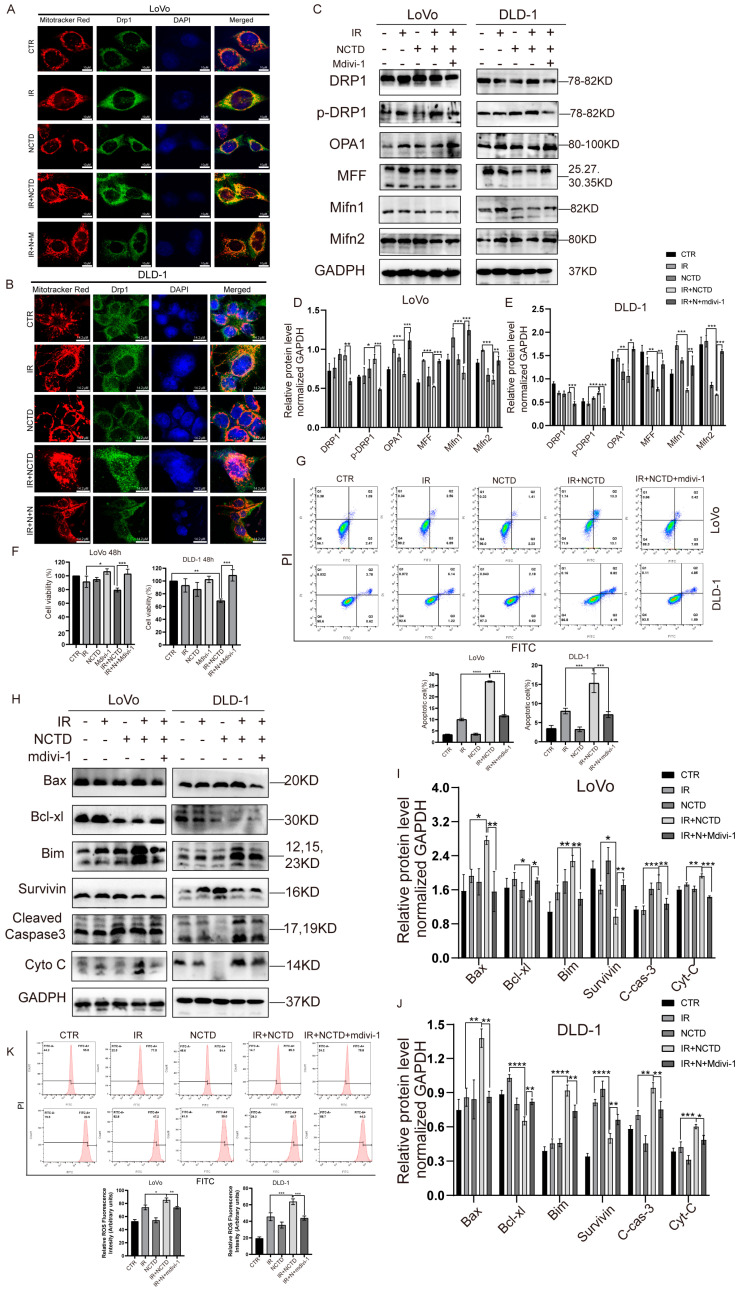
Mdivi-1 rescues mitochondria-dependent apoptosis and decreases ROS. (**A**,**B**) Immunofluorescence of DRP1 and mitochondria after Mdivi-1 (5 µmol/L) treatment in CRC cells. Scale bar = 10 µm. (**C**) Representative immunoblotting of mitochondrial fusion- and fission-associated proteins after Mdivi-1 treatment in CRC cells. (**D**,**E**) Statistical histogram of the relative expression of proteins associated with mitochondrial division and fusion after pretreatment with Mdivi-1. *, *p* < 0.05; **, *p* < 0.01, ***, *p* < 0.001. (**F**) Changes in cell viability after treatment with Mdivi-1 in CRC cells. *, *p* < 0.05; **, *p* < 0.01, ***, *p* < 0.001. (**G**) Apoptosis rate of CRC cells after Mdivi-1 treatment. ***, *p* < 0.001, ****, *p* < 0.0001. (**H**) Western blotting for Bax, Bim, Bcl-xl, survivin, cleaved caspase-3, and Cyt-C expression in CRC cells. (**I**,**J**) Histogram of relative expression of apoptotic proteins after Mdivi-1 pretreatment. *, *p* < 0.05; **, *p* < 0.01, ***, *p* < 0.001, **** *p* < 0.0001. (**K**) Changes in ROS after Mdivi-1 treatment in CRC cells. *, *p* < 0.05; **, *p* < 0.01, ***, *p* < 0.001.

**Figure 8 antioxidants-13-00347-f008:**
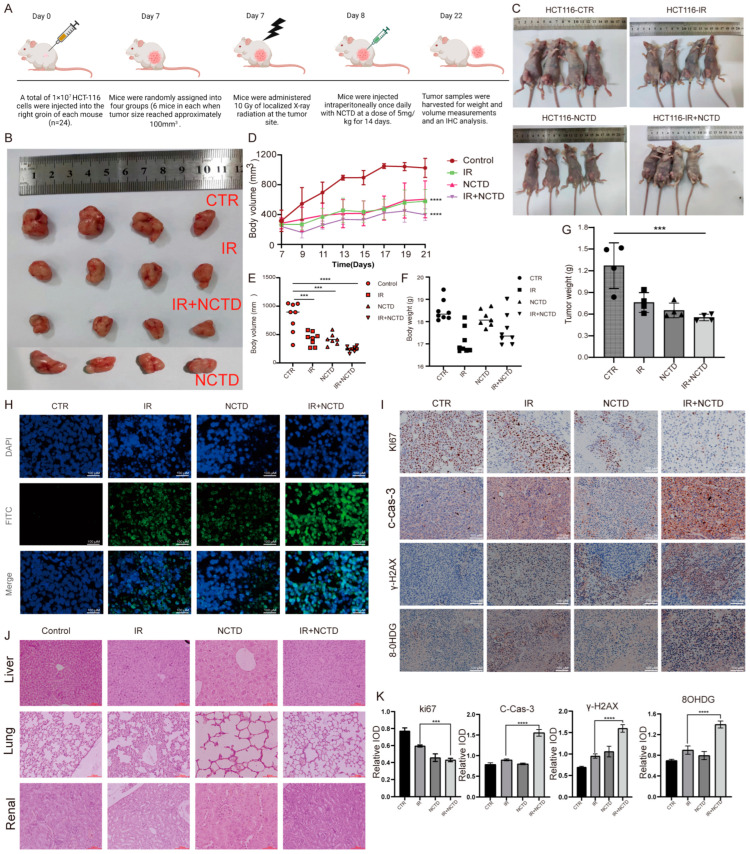
NCTD enhances CRC cell radiosensitivity in vivo. (**A**–**E**) Tumor growth curves and images of tumors in each group (n = 6 and 4 tumors were observed) ( ***, *p* < 0.001, **** *p* < 0.0001). (**F**) Body weight of mice in each group. (**G**) Quality of tumors in each group (*** *p* < 0.001). (**H**) Detection of apoptosis in mouse tumor tissues by TUNEL staining. (**I**,**K**) Immunohistochemical staining to test the expression of Ki67, cleaved caspase-3, γ-H2AX, and 8-OHDG in tumor sections from all study groups, ***, *p* < 0.001, **** *p* < 0.0001. (**J**) Hematoxylin and eosin staining of tumor sections from all groups to detect the effects of NCTD on the liver, lungs, and kidneys of mice.

**Figure 9 antioxidants-13-00347-f009:**
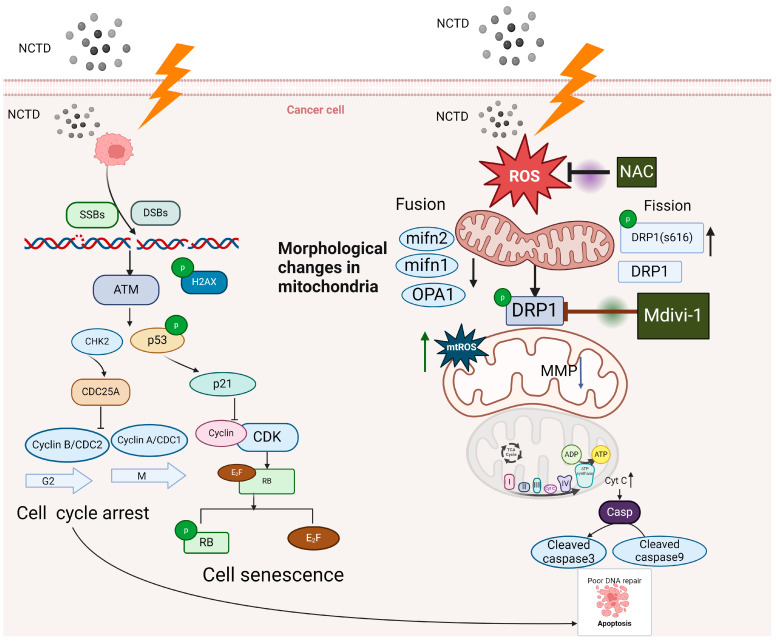
Mechanism of action of NCTD in increasing sensitivity to radiotherapy in colorectal cancer.

## Data Availability

All data generated in this study are included in this article and its Appendix A.

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
