# Peer review of "Norcantharidin Sensitizes Colorectal Cancer Cells to Radiotherapy via Reactive Oxygen Species–DRP1-Mediated Mitochondrial Damage"

_antioxidants, 2024, doi:10.3390/antiox13030347_

Round 1
Reviewer 1 Report
This study determined whether norcantharidin (NCTD) improves therapeutic efficacy in colorectal cancer when combined with radiotherapy. Additionally, the authors explained the mechanism of action of NCTD focusing on ROS-mediated mitochondrial damage. This study provides extensive data on the effects of NCTD through experiments at the cellular level and in animal models. However, much improvement is needed in the reliability of experimental results and the way they are presented.
1. The author should explain in more detail in the introduction or discussion section the previous studies on how NCTD induces apoptosis in cancer cells (prostate, gastric, breast cancer, etc., as described in the introduction). Moreover, the authors stated in the discussion section that NCTD is widely used in the clinical treatment of CRC (line 541-542). If so, I think we will now know in detail how NCTD affects CRC cells. Discussion on this should be added.
2. Why did the authors use IC20 rather than IC50 to determine drug concentration? Moreover, it is doubtful whether IC20 values of 10 and 50 μM in LoVo and DLD-1 cells, respectively, are calculated as integers. Please specify the calculation method or software used.
3. Additionally, it should be described whether the concentration of NCTD used in this study is related to the concentration used in clinical practice or the concentration used in previous studies.
4. The authors suggested that the SER of radiotherapy in LoVo and DLD-1 cells was 1.75 and 1.04, respectively. Why did the authors select 6 Gy, which is much higher than SER, as the appropriate dose?
5. The organization of all figures and the text of the figure legends should be reviewed more carefully and improved. To make it easier for readers to understand, the figures should be arranged sequentially according to the figure panels (A, B, C, D, ...) and the font size should be kept constant. Additionally, figure legends sometimes do not adequately describe each figure, or even omit them altogether. Authors must check carefully.
6. What is the difference between the results shown in Figures 1B-C and 1D-E? Were different methods used? Figures 1D-E are not shown in the main text.
7. Although it is marked '(K-P)' in the legend of Figure 1, this panel is not presented on the figure.
8. Why are CTR and IR+NCTD compared directly in Figures 1J-K
9. The quality of all Western Blot figures is poor. In particular, there are many inconsistencies between the intensity of the bands shown and the quantified graph. For example, in Figure 2C, can we say that the expression of γ-H2AX is higher in the IR-only or NCTD-only groups than in the CTR? This trend is prevalent in all Western blot images and graphs, and without improvement this study will not be suitable for publication. In addition, the background of the Western blot band also has many parts where the brightness difference varies sharply along the border, which also greatly lowers the image quality (e.g. Figure 2J-Bax, Survivin, GAPDH (DLD-1)).
10. What does 'increased mRNA levels of p-DRP1' mean? (line 389-390)
11. Why are DRP1 mRNA levels increased in the IR+NCTD group but protein levels are no different?
12. Figure legends 4A and 4B have been swapped.
13. What does it mean that cell division promotes apoptosis (line 578)? Please describe in more detail.
Author Response
Dear Editor
I am submitting a revision of the above referenced manuscript by Xu et al, entitled “Exploring the role and mechanism of norcantharidin on the sensitization of rectal cancer radiotherapy based on ROS-DRP1-mediated mitochondrial damage” (ID: antioxidants-2875298), for consideration for publication in antioxidants.
Thank you very much for your email on review comments. We appreciate your patience and valuable suggestions on our manuscript. We have carefully revised the manuscript according to the suggestions and guidelines. The revised manuscript has been uploaded to antioxidants online manuscript submission and tracking system. In this manuscript, the major modifications were noted with red font. According to the comments of reviewers, we will answer point-by-point response in the file of “Point-by-point response to reviewers”.
We thank the reviewers for their insightful comments and helpful suggestions that have helped us to strengthen the manuscript substantially. We believe that we have addressed the reviewers’ concerns fully and to the best of our ability. We hope the paper now is fit for publication in antioxidants.
Thank you very much for your consideration. I look forward to hearing from you favorably.
Best regards,
Project team members
Reviewer1:
- The author should explain in more detail in the introduction or discussion section the previous studies on how NCTD induces apoptosis in cancer cells (prostate, gastric, breast cancer, etc., as described in the introduction). Moreover, the authors stated in the discussion section that NCTD is widely used in the clinical treatment of CRC (line 541-542). If so, I think we will now know in detail how NCTD affects CRC cells. Discussion on this should be added.
- Why did the authors use IC20 rather than IC50 to determine drug concentration? Moreover, it is doubtful whether IC20 values of 10 and 50 μM in LoVo and DLD-1 cells, respectively, are calculated as integers. Please specify the calculation method or software used.
- Additionally, it should be described whether the concentration of NCTD used in this study is related to the concentration used in clinical practice or the concentration used in previous studies.
- The authors suggested that the SER of radiotherapy in LoVo and DLD-1 cells was 1.75 and 1.04, respectively. Why did the authors select 6 Gy, which is much higher than SER, as the appropriate dose?
- The organization of all figures and the text of the figure legends should be reviewed more carefully and improved. To make it easier for readers to understand, the figures should be arranged sequentially according to the figure panels (A, B, C, D, ...) and the font size should be kept constant. Additionally, figure legends sometimes do not adequately describe each figure, or even omit them altogether. Authors must check carefully.
- What is the difference between the results shown in Figures 1B-C and 1D-E? Were different methods used? Figures 1D-E are not shown in the main text.
- Although it is marked '(K-P)' in the legend of Figure 1, this panel is not presented on the figure.
- Why are CTR and IR+NCTD compared directly in Figures 1J-K
- The quality of all Western Blot figures is poor. In particular, there are many inconsistencies between the intensity of the bands shown and the quantified graph. For example, in Figure 2C, can we say that the expression of γ-H2AX is higher in the IR-only or NCTD-only groups than in the CTR? This trend is prevalent in all Western blot images and graphs, and without improvement this study will not be suitable for publication. In addition, the background of the Western blot band also has many parts where the brightness difference varies sharply along the border, which also greatly lowers the image quality (e.g. Figure 2J-Bax, Survivin, GAPDH (DLD-1)).
- What does 'increased mRNA levels of p-DRP1' mean? (line 389-390)
- Why are DRP1 mRNA levels increased in the IR+NCTD group but protein levels are no different?
- Figure legends 4A and 4B have been swapped.
- What does it mean that cell division promotes apoptosis (line 578)? Please describe in more detail.
Reply:
We sincerely thank the reviewer’s helpful suggestions. We do very much agree with you. As you can see, We have added the related content in the Introduction section to make the article more coherent,We have added the study of induction of apoptosis, primarily by the mitochondrial pathway and by direct induction of apoptosis.
The manuscript was modified as follows:
Page 2 Lines 83-90 of the Introduction as follows:
For example, Shen et al. showed that NCTD induces mitochondrial membrane potential (MMP) reduction and mitochondrial dysfunction and revealed that the accumulation of intercellular ROS can eventually induce apoptosis by causing DNA damage [18-20]. In addition, NCTD can directly affect the expression of Bax, Bcl-2, and caspase-3 and -9. Specifically, NCTD significantly increases the expression of caspase-3, -8, and -9, decreases the expression of caspase-4 and -12 and anti-apoptotic proteins Bcl-XL and Mcl-1, and increases the expression of the pro-apoptotic protein Bak.
In the discussion we added about how NCTD affects colorectal cancer cells, mainly by inducing cell death through inhibition of cell signaling and other pathways, as follows: NCTD has been shown to inhibit the expression of Ki-67 and Bcl-2, induce S-phase cell cycle arrest, and directly or indirectly downregulate the VEGF-A,-C,-D/VEGFR-2,-3 signaling pathways, thereby inhibiting the proliferation of CRC cells [34]. Qiu et al. showed that NCTD inhibits cell growth by inhibiting the expression and phosphorylation of EGFR and c-Met in human colon cancer cells. Another study showed that NCTD inhibits tumor angiogenesis by blocking the VEGFR2/MEK/ERK signaling pathway [35]. In addition, NCTD inhibits epithelial-mesenchymal transition in colon cancer cells by inhibiting the alphavbeta6-ERK-Ets1 signaling pathway [36]. However, few studies have focused on increased sensitivity of NCTD to radiotherapy in rectal cancer, and the mechanism underlying the ability of NCTD to overcome radiation tolerance in rectal cancer has not been reported. Therefore, we sought to determine whether NCTD could increase CRC radiotherapy sensitization by increasing ROS production.
.(Page 21 Lines 542-554)
Question 2
Reply: We appreciate your patience and valuable suggestions on our manuscript. In the study of sensitization by radiotherapy, the drug concentration of IC20 is generally considered to be the maximum dose that is ineffective in killing the cells, or defined as small and low doses, the radiosensitizer must meet the following conditions: The therapeutic dose is non-toxic or of low toxicity to normal cells and has little or no sensitizing effect on normal cells, and at the same time, in order to avoid the toxic effect of the drug itself on the cells, we refer to a large body of literature[1-5], so we chose IC20 as the concentration for the follow-up experiments. We used Graphpad prism9 software to calculate IC20, and the IC20 values of HCT-116 and DLD-1 were 9.455 and 50.467, respectively, and for the convenience of calculation and dosing, we took 10 and 50 as the subsequent drug concentrations. The calculations for IC20 have been added to the manuscript.(Page 6 Lines 289-262). In addition, the concentration of IC20 used in this study is not related to the concentration used in the clinic, and we have conducted several pre-experiments and CCK8 experiments to finalize the concentration of the drug, which is the inhibitory concentration of LoVo and DLD-1 in colorectal cancer cells, and we have also reviewed the literature and found that the concentration of NCTD is similar to our results in other cell lines[6].
Question 4
Reply: We really appreciate your patience and valuable suggestions on our manuscript. The effect of a radiosensitizer is often expressed as a sensitization ratio (SER), SER = D0 (without sensitizer)/D0 (with sensitizer). It is the ratio of the irradiation dose needed to achieve a specific biological effect when irradiated alone and the irradiation dose needed to achieve the same biological effect after irradiation combined with the application of a radiosensitizer; SER is greater than 1 to prove that it has a radiosensitizing effect, and we have confirmed that the NCTD is able to increase the sensitivity of colorectal cancer to radiosensitization, whereas the selection of the dose of radiotherapy is based on my pre-tests, the results of the pre-tests of the subject group, and the results of the CCK8 experiments, and we finally chose 6Gy as our exposure dose.
Question 5、6
Reply: We apologize for any inconvenience caused by our carelessness. I have carefully modified all the pictures, I have corrected and removed the unnecessary legends. Figure 1B-C is a line graph of the changes in cell viability of the 2 groups of cell lines after 24, 48, and 72 h of drug administration, with line graphs you can see the changes in cell viability over time more clearly and more intuitively, and Figure 1D-E is a bar graph of the changes in cell viability after 48 h of drug administration, and the bar graphs can be more intuitively seen as the changes in cell viability with the concentration of the drug, which I've labeled in the text, and also illustrated in the legend of the figure. Thanks very much for your attention to our paper.
Question 7
Reply: We apologize for mislabeling "H-K" as "K-P" in Figure 1, we have checked the legend carefully and we have corrected it. Thank you very much for your attention to our paper.
Question 8
Reply: We sincerely thank the reviewer’s insightful comments and helpful suggestions that have helped us to strengthen the manuscript substantially. We have made corrections based on your suggestions. We have re-graphed the graphs as per your request, comparing the IR group to the IR+NCTD group, and we have done the statistics again and found that there is a statistically significant difference between the IR group compared to the IR+NCTD grou(p<0.05).
Question 9
Reply: We deeply appreciate your review and valuable comments above. We strongly agree with your opinion. We have checked all the images to make the information more accessible to the readers. You proposed that in Figure 2C, the expression of γ-H2AX is higher in irradiation alone group and drug administration alone group than control group, in our study, we think that the expression of γ-H2AX in IR group is higher than control group because ionizing radiation can directly cause DNA damage, the difference is that NCTD alone can lead to the elevation or not of the expression of γ-H2AX, which is reasonable, because the The amount of our NCTD is IC20 value, which does not have a strong killing effect on the cells. And studies have also shown that NCTD can lead to DNA damage, and apoptotic cell cycle blockade[7, 8]. But the expression of γ-H2AX after IR combined with NCTD treatment is significantly higher than that of the irradiation group alone, which is consistent in all of my WB images.
For your question about the protein blotting image, our image is from my real experimental results, without any processing, may be because of the lack of experience at that time, the range of film cutting is small, the exposure for the parameters of the setting is not so perfect, I re-did the WB, a new image with better exposure was selected for correction and statistically reanalyzed the wb images, and the modifications are as follows:Figure 2, DLD-1 (Bax, Survivin, GAP、γ-H2AX), LoVo cell line (Bim、γ-H2AX), Figure 4 LoVo (Mifn2). In addition, you said that there is a gap between the statistical analysis and the image, this is because the statistical analysis is the result of 3 experiments, there may be a slight difference with the existing presentation of the image, these results are all the image I made with Graphpad prism9 software, I am very sorry for this, I will try my best to modify according to your requirements.
Question 10
We are very sorry for the mistakes in this manuscript and the inconvenience they caused in your reading. We have thoroughly checked and corrected the error labels that we found in our revised manuscript, we have corrected p-DRP1 to DRP1. Thanks so much for your useful comments.
Question 11
Response: We appreciate your valuable suggestions on our paper. The mRNA level of DRP1 was elevated, while the DRP1 ontology was not significantly elevated, which was the result of many experiments throughout my study, which is what I was puzzled about at the beginning, because it was not exactly the same as my WB protein result, so I performed many repetitions of the experiment, extracting the mRNA with different batches of cells, and got the same result in all the repetitions of the experiment, which we hypothesized may be due to the fact that the synthesis of the mRNA is earlier than the protein and is degraded more quickly, but we are selecting the same time to lift the mRNA and to extract the protein. In addition RNA can not represent the level of protein expression, protein expression can be regulated by transcription, but also by the level of translation, regulation at the level of translation may lead to protein expression and mRNA expression is not proportional, for example, many proteins and a large number of protein genes and a large amount of amplification, while the protein does not amplify, or the gene is very little protein but a large number of expression, we have seen that there is a literature of protein expression and gene expression does not coincide, as follows:
Regulation of XIAP translation and induction by MDM2 following irradiation[9].
Question 12
We are very sorry for the mistakes in this manuscript and the inconvenience they caused in your reading. We have thoroughly checked and corrected the error labels that we found in our revised manuscript, we have corrected Figure4A to Figure 4B. Thanks so much for your useful comments.
Question 13
I am very sorry that my inattentiveness led to this error, I sincerely hope you accept my apology, here it was supposed to be that mitochondrial division promotes the onset of apoptosis, I have corrected it in the original text, again I apologize, I will also revise the whole text carefully, I hope you can accept my revised text.
Reviewer 2 Report
In the article entitled “Exploring the role and mechanism of norcantharidin on the sensitization of rectal cancer radiotherapy based on ROS-DRP1-mediated mitochondrial damage”, the authors investigated the role and mechanism of action of norcantharidin (NCTD), a derivative of cantharidin, which is able to induce ROS generation and is widely used in the treatment of CRC, in overcome the radiation tolerance of CRC increasing the sensitivity to radiotherapy.
They treated different CRC cell lines with NCTD, ionizing radiation or a combination of the two and they found that NCTD significantly inhibited the proliferation of CRC cells and enhanced their sensitivity to radiotherapy. They analyzed the molecular mechanisms through which ionizing radiation (IR) combined with NCTD acted increasing sensitivity of CRC cells to radiotherapy.
They concluded that NCTD/IR combination would be used as a potential therapeutic approach for the treatment of CRC.
Data reported are interesting but they need to be deeply improved, in order to be considered for publication. I think the manuscript cannot be accepted in its present form and needs to be deeply revised. I reported here my comments:
The authors should add an ABBREVIATIONS Section to the manuscript in order to clarify the meaning of such acronyms reported within the text.
The authors should add a GRAPHICAL ABSTRACT to the manuscript in order to clarify the hypothesis and the results obtained.
In general, English style has to be verified due to many typing errors.
Many errors are evidenced in Tables/Figures list:
- In the MATERIALS AND METHODS Section, sub-paragraph named “2.6 Western blot analysis”, the authors reported that “The membrane was then incubated with the diluted antibody (Supplementary Material 1) overnight on a shaker at 4 ºC.”, but in the Suppl. Material, it is TABLE S2.
- In the MATERIALS AND METHODS Section, sub-paragraph named “2.12 RNA isolation and quantitative real-time”, the authors reported that “Gene expression was normalized to GADPH. The PCR primers (Sangon Biotech, Shanghai, China) are shown in Supplementary Material 2.” but in the Suppl. Material, it is TABLE S1. Again, within the title it is reported that shRNA sequences were reported in, but they don’t. Please correct these mistakes.
- FIGURES 1D-E, 2D-L, 4F, 5O-P, 6F-G, 7E and SUPPL. FIGURE S1 are not commented within the text. Please clarify that and add comments.
In the MATERIALS AND METHODS Section, sub-paragraph named “2.15 TUNEL assay”, the authors reported that “The tissue sections were routinely dewaxed and gradient alcohol hydration.”
The sentence seems to lack a verb. Please verify and eventually correct it.
What is the difference between the meaning of FIGURES 1B-C and 1D-E? Please clarify that.
What is the meaning of FIGURES 2D-E? Please clarify that.
In FIGURE 3I, the authors reported the immunoblotting of cell-cycle proteins cyclin B1, p-CDC2, p-CHK2. I think that they have to report also the levels of TOTAL CDC2 and CHK2 proteins, close to the phosphorylated forms, in order to confirm the results with respect to translationa or post-translational mechanisms.
In the RESULTS Section, sub-paragraph named “3.4 NCTD combined with IR impairs mitochondrial morphology and function leading to increase mitochondrial division”, the authors reported that “we used JC-1 to measure MMP.”. Please clarify the meaning of JC-1 and the mechanisms by which it works in the MATERIALS and METHODS Section.
In FIGURES 4C and 7F, please ad the histograms regarding proteins/GAPDH ratio.
Have the authors analyzed clinical samples or datasets in order to support their conclusions or to possible extend their analysis to other solid tumors?
In conclusion, data are potentially interesting but they need to be improved in order to be taken in consideration for publication.
The paper needs to be greatly improved in its content in order to meet the journal aims and be considered for publication.
It is all reported in my comments
Author Response
Dear Editor
I am submitting a revision of the above referenced manuscript by Xu et al, entitled “Exploring the role and mechanism of norcantharidin on the sensitization of rectal cancer radiotherapy based on ROS-DRP1-mediated mitochondrial damage” (ID: antioxidants-2875298), for consideration for publication in antioxidants.
Thank you very much for your email on review comments. We appreciate your patience and valuable suggestions on our manuscript. We have carefully revised the manuscript according to the suggestions and guidelines. The revised manuscript has been uploaded to antioxidants online manuscript submission and tracking system. In this manuscript, the major modifications were noted with red font. According to the comments of reviewers, we will answer point-by-point response in the file of “Point-by-point response to reviewers”.
We thank the reviewers for their insightful comments and helpful suggestions that have helped us to strengthen the manuscript substantially. We believe that we have addressed the reviewers’ concerns fully and to the best of our ability. We hope the paper now is fit for publication in antioxidants.
Thank you very much for your consideration. I look forward to hearing from you favorably.
Best regards,
Project team members
Reviewer 2:
The authors should add an ABBREVIATIONS Section to the manuscript in order to clarify the meaning of such acronyms reported within the text.
The authors should add a GRAPHICAL ABSTRACT to the manuscript in order to clarify the hypothesis and the results obtained.
In general, English style has to be verified due to many typing errors.
Many errors are evidenced in Tables/Figures list:
In the MATERIALS AND METHODS Section, sub-paragraph named “2.6 Western blot analysis”, the authors reported that “The membrane was then incubated with the diluted antibody (Supplementary Material 1) overnight on a shaker at 4 ºC.”, but in the Suppl. Material, it is TABLE S2.
In the MATERIALS AND METHODS Section, sub-paragraph named “2.12 RNA isolation and quantitative real-time”, the authors reported that “Gene expression was normalized to GADPH. The PCR primers (Sangon Biotech, Shanghai, China) are shown in Supplementary Material 2.” but in the Suppl. Material, it is TABLE S1. Again, within the title it is reported that shRNA sequences were reported in, but they don’t. Please correct these mistakes.
FIGURES 1D-E, 2D-L, 4F, 5O-P, 6F-G, 7E and SUPPL. FIGURE S1 are not commented within the text. Please clarify that and add comments.
In the MATERIALS AND METHODS Section, sub-paragraph named “2.15 TUNEL assay”, the authors reported that “The tissue sections were routinely dewaxed and gradient alcohol hydration.”
The sentence seems to lack a verb. Please verify and eventually correct it.
What is the difference between the meaning of FIGURES 1B-C and 1D-E? Please clarify that.
What is the meaning of FIGURES 2D-E? Please clarify that.
In FIGURE 3I, the authors reported the immunoblotting of cell-cycle proteins cyclin B1, p-CDC2, p-CHK2. I think that they have to report also the levels of TOTAL CDC2 and CHK2 proteins, close to the phosphorylated forms, in order to confirm the results with respect to translationa or post-translational mechanisms.
In the RESULTS Section, sub-paragraph named “3.4 NCTD combined with IR impairs mitochondrial morphology and function leading to increase mitochondrial division”, the authors reported that “we used JC-1 to measure MMP.”. Please clarify the meaning of JC-1 and the mechanisms by which it works in the MATERIALS and METHODS Section.
In FIGURES 4C and 7F, please ad the histograms regarding proteins/GAPDH ratio.
Have the authors analyzed clinical samples or datasets in order to support their conclusions or to possible extend their analysis to other solid tumors?
Reply:
- The authors should add an ABBREVIATIONS Section to the manuscript in order to clarify the meaning of such acronyms reported within the text.
Response: We deeply appreciate your review and valuable comments above. We strongly agree with your opinion. I've added acronyms to the text to make the information more accessible to the readers. If you have any better suggestions, please don't hesitate to let us know.
- The authors should add a GRAPHICAL ABSTRACT to the manuscript in order to clarify the hypothesis and the results obtained.
Response: We appreciate your valuable suggestions on our paper. I strongly agree with your comments. I've added graphic summaries to the text, thank you very much.
- In general, English style has to be verified due to many typing errors.
Response: We apologize for the typing errors of our manuscript. We worked on the manuscript for a long time and repeated addition and removal of sentences and sections obviously led to poor readability. We have now worked on both language and readability and have also involved native English speakers for language corrections. We really hope that the flow and language level have been substantially improved.
- In the MATERIALS AND METHODS Section, sub-paragraph named “6 Western blot analysis”, the authors reported that “The membrane was then incubated with the diluted antibody (Supplementary Material 1) overnight on a shaker at 4 ºC.”, but in the Suppl. Material, it is TABLE S2. In the MATERIALS AND METHODS Section, sub-paragraph named “2.12 RNA isolation and quantitative real-time”, the authors reported that “Gene expression was normalized to GADPH. The PCR primers (Sangon Biotech, Shanghai, China) are shown in Supplementary Material 2.” but in the Suppl. Material, it is TABLE S1. Again, within the title it is reported that shRNA sequences were reported in, but they don’t. Please correct these mistakes.
Response: We are very sorry for the mistakes in this manuscript and the inconvenience they caused in your reading. We have thoroughly checked and corrected the error labels that we found in our revised manuscript. Thanks so much for your useful comments.
- FIGURES 1D-E, 2D-L, 4F, 5O-P, 6F-G, 7E and SUPPL. FIGURE S1 are not commented within the text. Please clarify that and add comments.
Response: We appreciate your valuable suggestions on our paper. In order to make the figures 1 to 8 more clearly, we have added the textual annotations for every figures. Each chart is illustrated in detail. The modifications are as follows:
Figure 1. Figure 1. Norcantharidin (NCTD) enhances colorectal cancer (CRC) cell radiosensitivity in vitro. (A) Chemical structure of NCTD. (B-C) Viability of LoVo and DLD-1 cells treated with NCTD. (D-E) CRC cell viability and IC20 values after 48 h treatment with different concentrations of NCTD. (F-G) Colony formation assay showing enhanced radiosensitivity of CRC cells in vitro, as evaluated by the radiation multi-target single-hit model, after NCTD (10 and 50 µM) treatment. (H-K) CRC cells were treated with ionizing radiation (IR, 6 Gy) alone or in combination with NCTD (10 and 50 µM) for cell clone formation.
Figure 2. NCTD induces DNA damage and mitochondria-dependent apoptosis in CRC cells. (A–B) Fluorescence microscopy images showing increased DNA fracture (ϒ-H2AX, green) after the combination treatment of IR (6 Gy) and NCTD (10 and 50 µM). (C) Representative immunoblotting of DNA damage (ϒ-H2AX) in CRC cells. *, p < 0.05; **, p < 0.01. (D) Histogram of intracellular ϒ-H2AX-positive cytometry after IR combined with NCTD treatment. ****, p < 0.0001. (E) Casplab software was applied to process and analyze the comet images, and the lengths of the tail moments of the four groups of comets after different interventions were analyzed. **, p < 0.01. (F-G) Detection of DNA damage by comet assay. (H) NCTD increased apoptosis at 48 h post IR in CRC cells. (I) NCTD increased the expression of Cyt C 48 h post IR. (L) Histogram of Cyt C protein expression after different treatments. (J–K) Expression of Bcl-xl, Bax, Bim, cleaved caspase-3, and survivin was detected by western blotting 48 h post IR in CRC cells. *, p < 0.05; **, p < 0.01.
Figure 3. NCTD induces CRC cell senescence and blocks the cell cycle post IR. LoVo and DLD-1 cells were treated with NCTD for 24 or 48 h. (A–C) NCTD increased the expression of senescence-associated β-galactosidase (SA-β-gal) after IR in CRC cells. Scale bar = 100 µm. (D–F) Western blotting analysis of p21 and p16 in irradiated LoVo and DLD-1 cells. *, p < 0.05; **, p < 0.01. (G-H) CRC cell cycle was assessed by flow cytometry. (I) Representative immunoblotting of cell-cycle proteins cyclin B1、CDC2、 p-CDC2、CHK2 and p-CHK2. (J). Histogram of p21 and p16 protein gray values. *, p < 0.05; **, p < 0.01.
Figure 4. Effect of IR combined with NCTD on mitochondrial damage and mitochondrial division in CRC cells. (A) Morphology of mitochondria in CRC cells treated with IR and/or NCTD, as determined by confocal microscopy. Scale bar = 10 µm. (B) JC-1 in CRC cells. Scale bar = 100 µm. (C) Representative western blot results and quantification of mitochondrial fusion- and fission-associated proteins. **, p < 0.01. (D) JC-1 fluorescent probe bar chart, ****, p < 0.0001. (E-F) Statistical analysis of mRNA levels of mitochondrial fission- and fusion-associated proteins in CRC cells. *, p < 0.05; **, p < 0.01. (G-H) Representative confocal images showing DRP1 immunofluorescence and mitochondria stained with MitoTracker®. Nuclei were stained with DAPI. Scale bar = 10 µm. (I-J) Quantification of mitochondrial fusion- and fission-associated proteins. **, p < 0.01, ***, p < 0.001.
Figure 5. NCTD increased CRC cell apoptosis by upregulating ROS levels. (A–D) Combination therapy increased intracellular and mitochondrial ROS levels. (E) Images of clone formation in the groups after the addition of NAC. (F-G) NAC decreased intracellular ROS levels in CRC cells (**, p < 0.01). (H-J) NAC also inhibited ROS generation in mitochondria. *, p < 0.05; **, p < 0.01. (K–L) Detection of apoptosis by flow cytometry after Annexin V-propidium iodide (PI) staining. (M-N) Annexin V-PI staining histograms. (O-P) Representative immunoblotting of related apoptosis proteins and statistical histograms. *, p < 0.05; **, p < 0.01.
Figure 6. ROS lead to excessive mitochondrial division by increasing mitochondrial damage in CRC cells. (A–B) Mitochondrial membrane potential in CRC cells. *, p < 0.05; **, p < 0.01. (C) Mitochondrial morphology, as observed using confocal microscopy, NAC reverses mitochondrial fragmentation caused by combination therapy. Scale bar = 10 µm. (D) Representative immunoblotting of mitochondrial fusion and fission-associated proteins after NAC treatment in CRC cells. (E-F) Immunofluorescence of DRP1 and mitochondrial after NAC treatment in CRC cells. Scale bar = 10 µm. (G) Statistical bar graph of mitochondrial division and fusion. *, p < 0.05; **, p < 0.01.
Figure 7. Mdivi-1 rescues mitochondria-dependent apoptosis and decreases ROS. (A-B) Immunofluorescence of DRP1 and mitochondria after Mdivi-1 (5 µmol/L) treatment in CRC cells. Scale bar = 10 µm. (C) Representative immunoblotting of mitochondrial fusion- and fission-associated proteins after Mdivi-1 treatment in CRC cells. (D-E) Statistical histogram of the relative expression of proteins associated with mitochondrial division and fusion after pretreatment with Mdivi-1. *, p < 0.05; **, p < 0.01. (F) Changes in cell viability after treatment with Mdivi-1 in CRC cells. (G) Apoptosis rate of CRC cells after Mdivi-1 treatment. (H) Western blotting for Bax, Bim, Bcl-xl, survivin, cleaved caspase-3, and Cyt-C expression in CRC cells. (I-J) Histogram of relative expression of apoptotic proteins after Mdivi-1 pretreatment. *, p < 0.05; **, p < 0.01. (K) Changes in ROS after Mdivi-1 treatment in CRC cells. *, p < 0.05; **, p < 0.01.
- In the MATERIALS AND METHODS Section, sub-paragraph named “15 TUNEL assay”, the authors reported that “The tissue sections were routinely dewaxed and gradient alcohol hydration.”The sentence seems to lack a verb. Please verify and eventually correct it.
Response: Thank you very much for your seriousness and patience For our article, we have carefully reviewed the full text and asked native English speakers to revise our paper, thank you very much for your attention!
- What is the difference between the meaning of FIGURES 1B-C and 1D-E? Please clarify that. What is the meaning of FIGURES 2D-E? Please clarify that.
Response: We appreciate your valuable suggestions on our paper. I have carefully modified all the pictures, I have corrected and removed the unnecessary legends. Figure 1B-C is a line graph of the changes in cell viability of the 2 groups of cell lines after 24, 48, and 72 h of drug administration, with line graphs you can see the changes in cell viability over time more clearly and more intuitively, and Figure 1D-E is a bar graph of the changes in cell viability after 48 h of drug administration, and the bar graphs can be more intuitively seen as the changes in cell viability with the concentration of the drug, which I've labeled in the text, and also illustrated in the legend of the figure. Figure 2D shows the statistical histogram of γ-H2AX positive expression in the four groups after different treatments. Figure2E Casplab software was applied to process and analyze the comet images, and the lengths of the tail moments of the four groups of comets after different interventions were analyzed. **, p < 0.01. In addition, I've detailed this in the in-text legend. Thanks very much for your attention to our paper.
- In FIGURE 3I, the authors reported the immunoblotting of cell-cycle proteins cyclin B1, p-CDC2, p-CHK2. I think that they have to report also the levels of TOTAL CDC2 and CHK2 proteins, close to the phosphorylated forms, in order to confirm the results with respect to translationa or post-translational mechanisms.
Response: We deeply appreciate your review and valuable comments above. We strongly agree with your opinion. Since the relevant antibodies were not available in our laboratory, we repurchased the antibodies and performed WB experiments, the results of which have been added to the figure3. Thanks very much for your attention to our paper.
- In the RESULTS Section, sub-paragraph named“4 NCTD combined with IR impairs mitochondrial morphology and function leading to increase mitochondrial division”, the authors reported that “we used JC-1 to measure MMP.”. Please clarify the meaning of JC-1 and the mechanisms by which it works in the MATERIALS and METHODS Section.
Response: Thank you very much for your reminder. I am very sorry that I didn't write clearly in the text, JC-1 is an experiment used to detect the mitochondrial membrane potential, the experimental procedure of JC-1 has been described in the Materials and Methods 2.11, and I added the working mechanism of JC-1 in the text, as follows:
In healthy cells, JC-1 monomers aggregate to form polymers and mitochondria show strong red fluorescence (excitation at 550 nm, emission at 600 nm). In apoptotic or necrotic cells, JC-1 exists as a monomer and mitochondria show strong green fluorescence (excitation wavelength 485 nm, emission wavelength 535 nm). The ratio of the red fluorescen
- In FIGURES 4C and 7F,please ad the histograms regarding proteins/GAPDH ratio.
Response: We really appreciate your patience and valuable suggestions on our manuscript. I couldn't agree more with your advice. We statistically analyzed the WB images of FIGURE4C and 7F, added in Figure 7.
11.Have the authors analyzed clinical samples or datasets in order to support their conclusions or to possible extend their analysis to other solid tumors?
Response: We sincerely thank the reviewer’s insightful comments and helpful suggestions that have helped us to strengthen the manuscript substantially. We are very sorry that we have not performed the relevant clinical data analysis yet, but we think your suggestions are very reasonable, constructive and inspiring, and our next work will be to start analyzing the clinical samples and datasets in order to further validate our results, and we hope that the results from our group will be applied to other solid tumors.
We sincerely thank you for their reviews and comments. We learn a lot and have rewritten the respective part according to the reviewer’s suggestions. The new manuscript has been re-submitted, in which the amended sentences are highlighted in red color. We especially like this journal and hope our revised manuscript can be accepted for publication. We will do our best to refine our manuscript.
参考文献
[1] Deng T, Zhu Q, Xie L, et al. Norcantharidin promotes cancer radiosensitization through Cullin1 neddylation-mediated CDC6 protein degradation[J]. Mol Carcinog, 2022,61(8):812-824.
[2] Zou Z W, Liu T, Li Y, et al. Melatonin suppresses thyroid cancer growth and overcomes radioresistance via inhibition of p65 phosphorylation and induction of ROS[J]. Redox Biol, 2018,16:226-236.
[3] Peng F, Wei Y Q, Tian L, et al. Induction of apoptosis by norcantharidin in human colorectal carcinoma cell lines: involvement of the CD95 receptor/ligand[J]. J Cancer Res Clin Oncol, 2002,128(4):223-230.
[4] Lei Y, Li H X, Jin W S, et al. The radiosensitizing effect of Paeonol on lung adenocarcinoma by augmentation of radiation-induced apoptosis and inhibition of the PI3K/Akt pathway[J]. Int J Radiat Biol, 2013,89(12):1079-1086.
[5] Sidhu H, Capalash N. Synergistic anti-cancer action of salicylic acid and cisplatin on HeLa cells elucidated by network pharmacology and in vitro analysis[J]. Life Sci, 2021,282:119802.
[6] Lin C L, Chen C M, Lin C L, et al. Norcantharidin induces mitochondrial-dependent apoptosis through Mcl-1 inhibition in human prostate cancer cells[J]. Biochim Biophys Acta Mol Cell Res, 2017,1864(10):1867-1876.
[7] Yu C C, Ko F Y, Yu C S, et al. Norcantharidin triggers cell death and DNA damage through S-phase arrest and ROS-modulated apoptotic pathways in TSGH 8301 human urinary bladder carcinoma cells[J]. Int J Oncol, 2012,41(3):1050-1060.
[8] Chen S, Wan P, Ding W, et al. Norcantharidin inhibits DNA replication and induces mitotic catastrophe by degrading initiation protein Cdc6[J]. Int J Mol Med, 2013,32(1):43-50.
[9] Gu L, Zhu N, Zhang H, et al. Regulation of XIAP translation and induction by MDM2 following irradiation[J]. Cancer Cell, 2009,15(5):363-375.
Round 2
Reviewer 1 Report
This study determined whether norcantharidin (NCTD) improves therapeutic efficacy in colorectal cancer when combined with radiotherapy. . This study provides extensive data on the effects of NCTD through experiments at the cellular level and in animal models.
The authors responded appropriately to all comments, and the manuscript was sufficiently improved.
Reviewer 2 Report
In the 2nd version of the article entitled “Norcantharidin sensitizes colorectal cancer cells to radiotherapy via ROS-DRP1-mediated mitochondrial damage”, the authors replied to all my questions.
With respect to my comments, they added many interesting data to the manuscript and the paper results more clear than in the first version, definitely. I really appreciate very much their effort in answering my questions.
I think the manuscript may be considered for publication in its present form.
No comments